# A microfluidic-induced *C. elegans* sleep state

Daniel L. Gonzales [ID] [1,2], Jasmine Zhou[3], Bo Fan[2] & Jacob T. Robinson[1,2,3,4]*

An important feature of animal behavior is the ability to switch rapidly between activity states, however, how the brain regulates these spontaneous transitions based on the animal's perceived environment is not well understood. Here we show a *C. elegans* sleep-like state on a scalable platform that enables simultaneous control of multiple environmental factors including temperature, mechanical stress, and food availability. This brief quiescent state, which we refer to as microfluidic-induced sleep, occurs spontaneously in microfluidic chambers, which allows us to track animal movement and perform whole-brain imaging. With these capabilities, we establish that microfluidic-induced sleep meets the behavioral requirements of *C. elegans* sleep and depends on multiple factors, such as satiety and temperature. Additionally, we show that *C. elegans* sleep can be induced through mechanosensory pathways. Together, these results establish a model system for studying how animals process multiple sensory pathways to regulate behavioral states.

---

[1] Applied Physics Program, Rice University, 6100 Main St., Houston, TX 77005, USA. [2] Department of Electrical and Computer Engineering, Rice University, 6100 Main St., Houston, TX 77005, USA. [3] Department of Bioengineering, Rice University, 6100 Main St., Houston, TX 77005, USA. [4] Department of Neuroscience, Baylor College of Medicine, One Baylor Plaza, Houston, TX 77030, USA. *email: jtrobinson@rice.edu

Understanding how animals select behaviors based on sensory information is a fundamental goal of neuroscience[1–3]; however, sensorimotor transformations can vary dramatically depending on the state of the animal's nervous system[4–7]. Wakefulness and arousal[8], locomotor activity states[9], satiety[10], attention[11], and emotions[4,12] represent a spectrum of physiological and neural states that can dramatically affect how animals respond to a given stimuli. Small animals like the nematode *C. elegans* are tractable model organisms for understanding how physiological and neural states combine with information from multiple sensory pathways and give rise to specific behavior[3,5,7,10,13,14].

Sleep is one example of a neural state that dramatically alters an animal's sensorimotor transformations[15,16]. Studies of sleep across the phylogenetic tree have shown that sensory systems transition to a reduced activity state that leads to a decreased animal response to external stimuli[17–21]. In addition, sleep and wakefulness have been shown to correspond with distinct patterns of neural activity in humans[15,16], rodents[16], and fruit flies[22]. Likewise, whole-brain recordings from *C. elegans* suggest that the majority of worm brain activity can be represented on a low-dimensional manifold[23], and that the activity on this manifold shifts from phasic to fixed-point attractor dynamics during developmental sleep[24]. In addition, studies with *C. elegans* have revealed molecular pathways[18,25–28], neural circuits[25,29–34], and neuropeptides[30,33,35,36] that drive nematode sleep and arousal, and some of these mechanisms are conserved in other animals[37–39]. These reports have paved the way for using *C. elegans* sleep as a model system to understand spontaneous brain-state transitions between sleep and wakefulness. Given the number of unique regulators of *C. elegans* sleep, there is a need to identify a behavior that facilitates our understanding of how brain-wide neural circuits transduce multiple external and internal factors to drive sleep-wake transitions. Here, we describe a *C. elegans* sleep behavior in a microfluidic environment, termed microfluidic-induced sleep, that is amenable to whole-brain imaging and is regulated both by the animal's physiological state and external sensory queues. While microfluidic-induced sleep is likely a form of previously reported nematode quiescence behaviors, such as stress-induced sleep and episodic swimming[31,40], the rapid rate of state transitions and the fact that these quiescence behaviors occur in microfluidic devices provides a number of advantages for studying the mechanisms of state transitions with precise environmental controllability, high-throughput screenings, and whole-brain imaging.

## Results

### Spontaneous *C. elegans* quiescent bouts in microfluidics.
When confined to microfluidic chambers we found that adult *C. elegans* rapidly and spontaneously switch between normal activity and brief quiescent bouts without any additional stimuli (Fig. 1a). We initially observed this behavior when immobilizing worms for recording body-wall muscle electrophysiology[41]. In these electrical recordings we observed minutes-long periods of muscle inactivity that corresponded to whole-animal quiescence[41]. We have since found that this quiescence occurs in a wide range of microfluidic geometries (Supplementary Movie 1, 2).

We found that the onset, frequency, and duration of quiescent bouts in microfluidic chambers are unique compared with previously reported sleep behaviors in adult *C. elegans* (Fig. 1). While larval *C. elegans* display quiescence during lethargus[18], quiescence in adult worms occurs in only a few situations, such as after several hours of swimming[40,42–44] or after exposure to extreme environmental conditions[31,35,36,45,46]. To determine how microfluidic-induced quiescence compares with the well-established episodic behavior that occurs in swimming animals and stress-induced quiescence that occurs from a 30 min heat shock, we quantified these behaviors using the same methods to analyze microfluidic-induced sleep (Fig. 1b–f). During a 4 h imaging period, we found that animals partially immobilized in microfluidic channels (50 μm width, ~0.004 μL per chamber, Supplementary Movie 2) displayed short (1.3 ± 0.1 min, mean ± sem, Fig. 1e) bouts that, on average, began within the first hour of imaging (51 ± 4 min, mean ± sem, Fig. 1d). Likewise, we also observed frequent, but longer (2.4 ± 0.3 min, mean ± sem, Fig. 1e) quiescent bouts when the microfluidic chambers were large enough to allow animals to swim (500 μm width, ~0.1 μL per chamber, Supplementary Movie 1). We compared these data with swimming-induced quiescence, where animals alternate between swimming and quiescence in a large multi-well device called WorMotel (Fig. 1b, 8 μL of buffer per well)[47]. In WorMotel, we recorded 5–7 times less sleep over the imaging period compared with microfluidic chambers (Fig. 1c) and longer quiescent bouts compared with either microfluidic geometry (3.6 ± 0.3 min, mean ± sem, Fig. 1e). In addition, the average quiescence onset time was more than double what we observed for microfluidic-induced sleep (175 ± 6 min, mean ± sem, Fig. 1d). For our final comparison point, we heat-shocked animals in the large microfluidic device for 30 min at 30 °C to induce stress-quiescence, then analyzed animal behavior during the heat shock plus a 30 min recovery period[31] (Fig. 1b). As expected, during the noxious heat stimulus animals displayed their first quiescent bout 4–6 times faster than in microfluidic chambers with no external heat applied (13 ± 0.7 min, mean ± sem, Fig. 1d).

These initial data show that compared with episodic swimming in WorMotel, microfluidic-induced sleep occurs with a higher frequency, faster onset and in shorter bouts (Fig. 1c–e). Similarly, microfluidic-induced sleep onset occurs on a significantly different timescale compared with stress-induced sleep (Fig. 1b, d). It is important to note that while the microfluidic-induced sleep phenotype shows a number of quantitative differences when compared with episodic swimming and stress-induced quiescence, it is not clear if microfluidic-induced quiescence can be classified as a new *C. elegans* behavior. It is possible that the microfluidic environment accelerates episodic swimming or introduces mild stressors that more slowly lead to stress-induced quiescence. Thus, we consider microfluidic-induced quiescence to merely display quantifiably distinct dynamics compared with other reported quiescent behaviors in *C. elegans* adults and is not necessarily a new behavioral state.

### Microfluidic-induced quiescence is a sleep state.
Our observation of spontaneous *C. elegans* quiescence in microfluidic chambers led us to determine whether this behavioral state transition meets the evolutionarily conserved criteria for classification as sleep: reversibility, a decreased response to stimuli, homeostasis, and a stereotypical posture[17]. *C. elegans* developmentally timed sleep as larvae meets all requirements[18,19,37,48–53]. However, reports of sleep in adult worms have only observed reversibility and a decreased response to stimuli[31,37]. Here, we tested for a stereotyped posture, reversibility, decreased response to stimuli, and a homeostatic rebound.

Similar to previous reports[48,50,51], we found that animals exhibited a stereotyped posture during quiescence (Fig. 2a). The body curvature of animals crawling on agar typically decreases during developmental sleep[48]. In our case, we found that animals swimming in microfluidics show increased body curvature during sleep (from 3.9 ± 0.04 radians during wakefulness to 4.5 ± 0.1 radians during quiescence (mean ± sem), $p < 0.0001$, Fig. 2a, right). This result is consistent with the fact that *C. elegans*

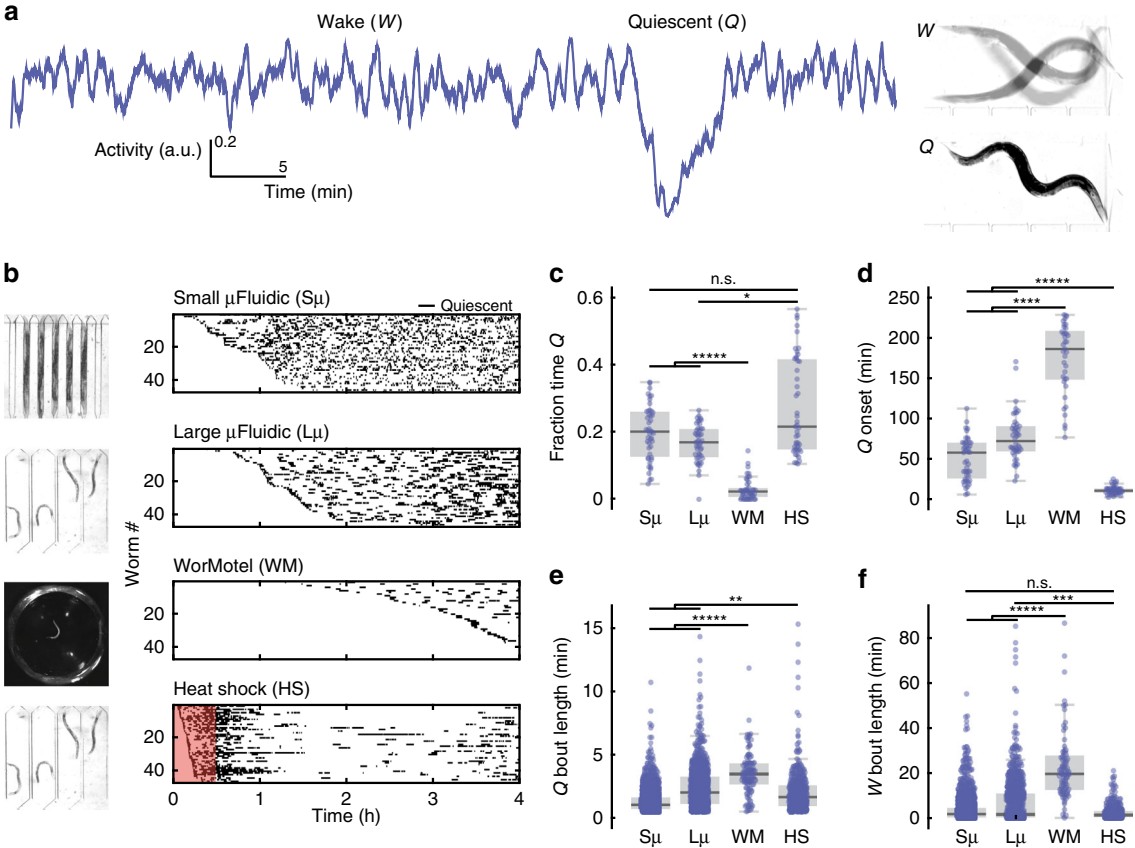

**Fig. 1** Quiescence dynamics are strongly affected by microfluidic environments. **a** Characteristic worm quiescence in a microfluidic chamber. *(Left)* 1-h activity trace from a worm swimming in a microfluidic chamber, calculated by subtracting consecutive frames to quantify movement (see "Methods" section). Quiescence is hallmarked by a clear drop in animal activity to near zero. *(Right)* 1-s overlap of frames shows the animal swimming during the wake period (W) and immobile during quiescence (Q). **b** All quiescent bouts recorded from animals in a small microfluidic chamber (50 μm width), large microfluidic chamber (500 μm width), WorMotel multi-well PDMS device, and large microfluidic device during a 30 min heat shock at 30 °C (see inset images). Raster plots show the bouts recorded from each animal during a 4 h imaging period (*n* = 47 animals for each condition). Experiments represented in the raster plots are sorted by the onset of the first detected sleep bout. **c–f** Quantitative sleep metrics. Sμ small microfluidic chambers, Lμ large microfluidic chambers, WM WorMotel, HS first hour of heat shock. **c** The total fraction of time each animal spent in the quiescent state. **d** Onset time of the first quiescent bout. **e** Length of individual quiescent bouts. **f** Length of individual wake bouts, excluding the first period of wake after animals are loaded into the chamber. (*n* = 47 animals for each condition; *p < 0.05, **p < 0.01, ***p < 0.001, ****p < 0.0001, *****p < 0.00001; Kruskal–Wallis with a post hoc Dunn–Sidak test). Source data is available as a Source Data file

quiescent posture, a hockey-stick-like shape[50], has a higher curvature than the long body wavelengths displayed during swimming locomotion[40]. To test for reversibility, we used blue light (5-s pulse, 5 mW/cm2) as a strong stimulus and found that this rapidly and reliably reversed the quiescent state, leading to a dramatic increase in both behavioral activity and nose speed (Fig. 2b, Supplementary Movie 3). To test for a decreased response to stimuli, we fabricated microfluidic push-down valves designed to deliver tunable mechanical stimuli. We found that when we applied a strong stimulus (high pressure, 30 psi), both wake and quiescent animals robustly responded with a significant increase in behavioral activity, which provides additional confirmation that the quiescent state is reversible (Fig. 2b, Supplementary Movie 4). When we applied a weak stimulus to wake animals (low pressure, 15 psi), we again reliably recorded a significant increase in activity that matched the response of the strong stimulation (Fig. 2c, Supplementary Movie 5). However, when we delivered a weak stimulus to quiescent animals, they responded weakly, exhibiting an average behavioral activity less than half that of animals in the awake state (Fig. 2c–d, Supplementary Movie 5). In fact, <40% of quiescent animals that received a weak stimulus transitioned to wakefulness,

compared with >75% of animals being in the awake state following stimulation for all other experimental conditions (Fig. 2e). Importantly, because the strong stimuli evoked a similar strong behavioral response from both quiescent animals and awake animals, animals in the quiescent state are not less capable of responding to stimuli provided the stimuli is sufficiently intense (Fig. 2c–d). Further analyses also showed that animals in the sleep state are less likely to respond to weak stimuli and transition to wakefulness, compared with animals in the wake state regardless of their activity level prior to the stimulus (Supplementary Fig. 1). When we analyzed data from the wake animals, we found that nearly 70% of animals with activity levels below our sleep threshold for 3 s prior to the weak stimuli transitioned to wakefulness, compared with <40% for animals that we classified to be in the sleep state prior to a weak stimulus (Supplementary Fig 1). These results are consistent with quiescence being a sleep state rather than simply a low-activity state of wakefulness[18,19,31,52].

In addition, we also tested for homeostatic rebound, which is found in worm developmental sleep[18,34,48]. Similar to previous studies with *C. elegans* developmental sleep[48], we hypothesized that longer wake bouts would lead to longer sleep bouts due to

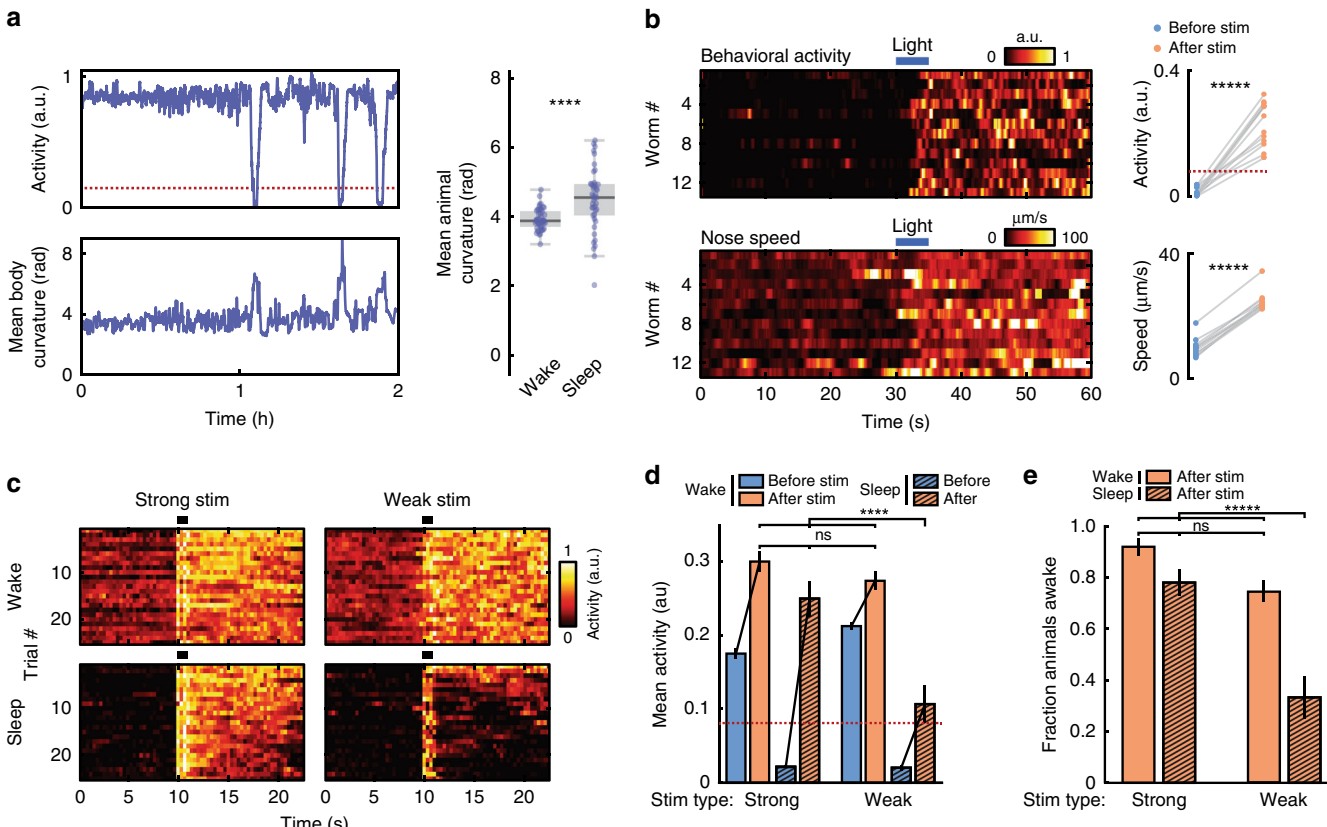

**Fig. 2** Sleep states show stereotypical posture, reversibility, and a decreased response to weak stimuli. **a** *C. elegans* show increased body curvature during sleep. (Left) Example of behavioral activity and normalized body curvature for a single animal. Dotted line indicates sleep threshold. (Right) Average body curvature during wakefulness and sleep across all animals (from Large microfluidic data in Fig. 1, $n = 47$ animals, ****$p < 0.0001$, unpaired $t$-test). **b** The sleep state is reversible. (Left) Heatmaps of behavioral activity (top) and nose speed (bottom) for quiescent animals that received a 5 s light pulse at $t = 30$ s. (Right) Mean behavioral activity (top) and nose speed (bottom) of each animal before and after stimulation. Dotted line indicated sleep threshold. ($n = 13$ worms, *****$p < 0.00001$, paired $t$-test). **c–e** Quiescent animals have a decreased response to weak sensory stimuli. **c** Heatmaps of activity from awake and sleeping animals that received strong or weak mechanical stimuli from microfluidic valves (Supplementary Movie 4, 5). For each condition, only the 25 trials with the highest mean activity post stimulation are shown. "Wake" indicates trials in which animals were awake but have a below-average activity before stimulation. "Sleep" indicates trials in which animals were quiescent before stimulation. Heatmaps contain an ~2 s long stimulation artifact beginning at 10 s due to movement from microfluidic valves. **d** Mean behavioral activity before and after mechanical stimulation from the trials in **c**. Dotted line indicates sleep threshold. In all cases, average activity significantly increases after the stimulation (largest $p$-value = 0.002, paired two-sided $t$-test). However, the behavioral activity after stimulation is significantly different when quiescent animals received weak mechanical stimuli (Error bars are sem; Strong-Wake $n = 108$, Strong-Sleep $n = 51$, Weak-Wake $n = 176$, Weak-Sleep $n = 29$; ****$p < 0.0001$, ns not significant, Kruskal–Wallis with a post hoc Dunn–Sidak test). **e** Fraction of animals awake following mechanical stimuli (Error bars are standard deviation, calculated by bootstrapping each data set with 5000 iterations; ns not significant, *****$p < 0.00001$; significance was calculated by data resampling 5000 iterations and a post hoc Bonferroni correction). Source data is available as a Source Data file

micro-homeostatic mechanisms. Indeed, we found that sleep bouts increased from $1.5 \pm 0.1$ min to $2.5 \pm 0.1$ min (mean ± sem) as the preceding wake bout increased in length from <1 min to 20 min (Supplementary Fig. 2A). However, even when wake bout lengths increased to longer than an hour, the sleep bouts lengths plateaued to an average of only $2.3 \pm 0.1$ min, contradicting the hypothesis that extended wake periods lead to sleep deprivation and homeostatic rebound. To directly test the effects of sleep deprivation, we performed optogenetic inhibition of the RIS neuron (Supplementary Fig 2B–D), which is known to be implicated in *C. elegans* sleep[29,30,34]. This optogenetic approach allowed us to have a control group (raised in the absence of all-trans retinal) that is also exposed to light but does not have optogenetic inhibition of the RIS neuron, thus controlling for any potential effects of stress produced by light illumination. We found that although RIS inhibition for 30 min did not fully abolish sleep, optically inhibited animals exhibited 324% more

sleep and 230% more low-activity behavior compared with control animals during the refractory period following the optogenetic inhibition of the RIS neuron ($p < 0.01$) (Supplementary Fig. 2D). RIS-inhibited animals also showed sleep bouts lasting 1.5 times longer than control animals ($p < 0.01$) (Supplementary Fig 2C–D). From our results combined, we conclude that animals show a homeostatic rebound in response to prolonged RIS inhibition. Therefore, microfluidic-induced quiescence indeed meets the behavioral precedents to be called a *C. elegans* sleep state: stereotyped posture, reversibility, a decreased response to stimuli, and homeostasis.

To further validate that microfluidic-induced sleep is a sleep state, we interrogated the role of two interneurons, RIS and ALA, known to dominate *C. elegans* developmental and stress-induced sleep, respectively, via at least partially distinct signaling pathways[37,54]. To test whether microfluidic-induced sleep is also dependent on these neurons, we compared the sleep phenotype of

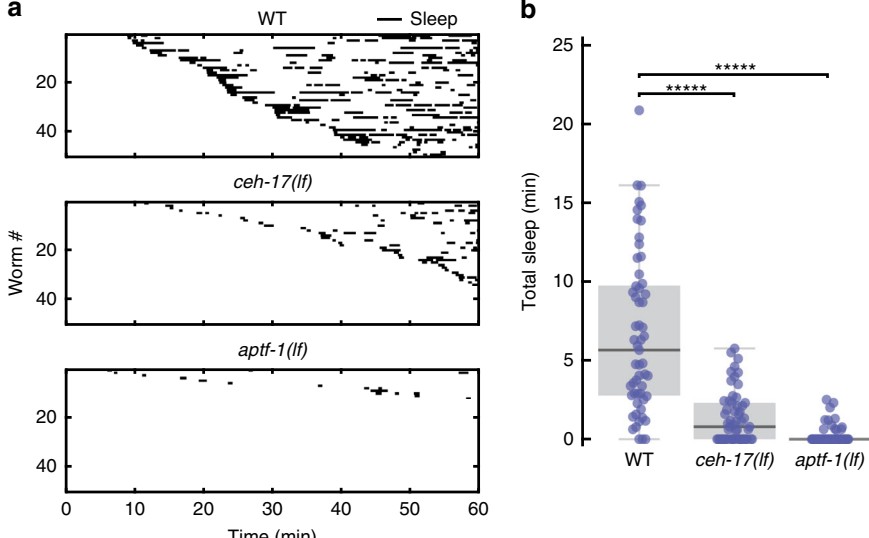

**Fig. 3** Microfluidic-induced sleep behavior depends on ALA and RIS neurons. **a** Raster plots of detected sleep bouts from WT, *ceh-17(lf)* and *aptf-1(lf)* animals. Only the top 50 animals showing the most total microfluidic-induced sleep are shown. **b** Both *ceh-17(lf)* and *aptf-1(lf)* show less total sleep than WT. The data suggest that microfluidic-induced sleep is strongly dependent on the ALA and RIS neurons. (WT $n = 57$, *ceh-17(lf)* $n = 57$, *aptf-1(lf)* $n = 60$; *****$p < 0.00001$ compared with WT, Kruskal–Wallis with a post hoc Dunn–Sidak test). Source data is available as a Source Data file

WT, *ceh-17(lf)*, and *aptf-1(lf)* loss-of-function mutants (Fig. 3). *ceh-17(lf)* shows defective axonal growth of the ALA neuron[26] and less stress-induced sleep[31]. *aptf-1(lf)* lacks the transcription factor necessary for RIS to signal quiescence via the FLP-11 neuropeptide and shows defects in developmental sleep[29,30]. In our experiments, both *ceh-17(lf)* and *aptf-1(lf)* mutants showed more than 4.5 times less total microfluidic-induced sleep than WT animals (Fig. 3b). Furthermore, while these behavioral data only represent sleep bouts lasting longer than 30 s to reduce false detections (see "Methods" section), these results—and others throughout this report—were not dependent on this choice of analyses (Supplementary Fig. 3). Together, these data validate that microfluidic-induced sleep is a *C. elegans* sleep state controlled by previously reported neural mechanisms.

**A global brain-state underlies microfluidic-induced sleep.** Having established that microfluidic-induced sleep meets the criteria for *C. elegans* sleep and is hallmarked by a dramatic behavioral state transition, we also sought to confirm that these spontaneous behavioral transitions were accompanied by an underlying global brain-state transition[24,55]. Previous work in chemically paralyzed animals used brain-wide calcium imaging to establish that the *C. elegans* nervous system, with the exception of a few neurons, transitions to a large-scale downregulation of neural activity during sleep[24,55].

Using whole-brain calcium-sensitive imaging we were able to establish that microfluidic-induced sleep is indeed associated with a global brain state transition (Fig. 4). The major advantage of the microfluidic-induced sleep behavior for whole-brain imaging is that animals can be confined in microfluidic chambers, which facilitates imaging without the use of chemical paralytics that abolish behavioral outputs[23,24,55]. Furthermore, animal confinement forgoes the need to perform real-time tracking during whole-brain imaging[56,57]. To exploit this advantage, we developed an imaging protocol using a microfluidic chamber geometry similar to previous studies[23] that partially immobilizes animals but allows for enough movement to quantify animal behavior and detect microfluidic-induced sleep during whole-brain calcium imaging (Fig. 4). By incorporating a period of animal habituation to blue excitation light, we were able to image continuously for

10 min using single-plane epifluorescence microscopy and capture spontaneous sleep-wake transitions in ~25% of animals (see Methods). Imaging a single 2D-plane allowed for quantifying both average ganglia activity and the activity of several individual neurons (Fig. 4b, Supplementary Movie 6).

With this imaging protocol, we observed a distinct correlation between animal behavior and neural activity (Fig. 4b, Supplementary Movie 6). During microfluidic-induced sleep both the average ganglia activity and the activity of individual neurons dropped significantly ($78 \pm 10\%$ and $55 \pm 10\%$, respectively) (Fig. 4c). However, as expected from previous work[24,29,30,34,55], we observed that some neurons actually increased in activity during microfluidic-induced sleep (Fig. 4c), some of which corresponded with the approximate location of the RIS interneuron (indicated neuron in Fig. 4a–b), which has been proposed as a sleep-promoting neuron for multiple *C. elegans* sleep-like states[29,30,34]. Single-neuron imaging confirmed that the RIS neuron indeed showed more than a twofold increase in calcium activity during microfluidic-induced sleep and strongly correlated with low-behavioral activity (Fig. 5). These results further support the claim that microfluidic-induced sleep is a *C. elegans* sleep behavior controlled by sleep-promoting circuits and corroborate previous reports that a unique brain state governs *C. elegans* sleep[24,55].

**Satiety, thermosensation and mechanosensation regulate sleep.** While the combination of behavioral and calcium-imaging data shows that microfluidic-induced sleep is hallmarked by a spontaneous brain and behavioral state transition (Fig. 4) regulated by sleep-promoting neurons like RIS and ALA (Fig. 3, Fig. 5), we also wanted to understand what neural circuits upstream of these neurons are involved in regulating microfluidic-induced sleep. Based on our observation of distinct phenotypes when comparing swimming animals in WorMotel to swimming animals in microfluidic chambers (Fig. 1), we hypothesized that *C. elegans* sensory circuits detect features unique to the microfluidic environment. To elucidate the cues regulating microfluidic-induced sleep, we used the versatility of microfluidics to perform assays under a variety of conditions and compared the microfluidic-induced sleep phenotype to the baseline behavior observed in a

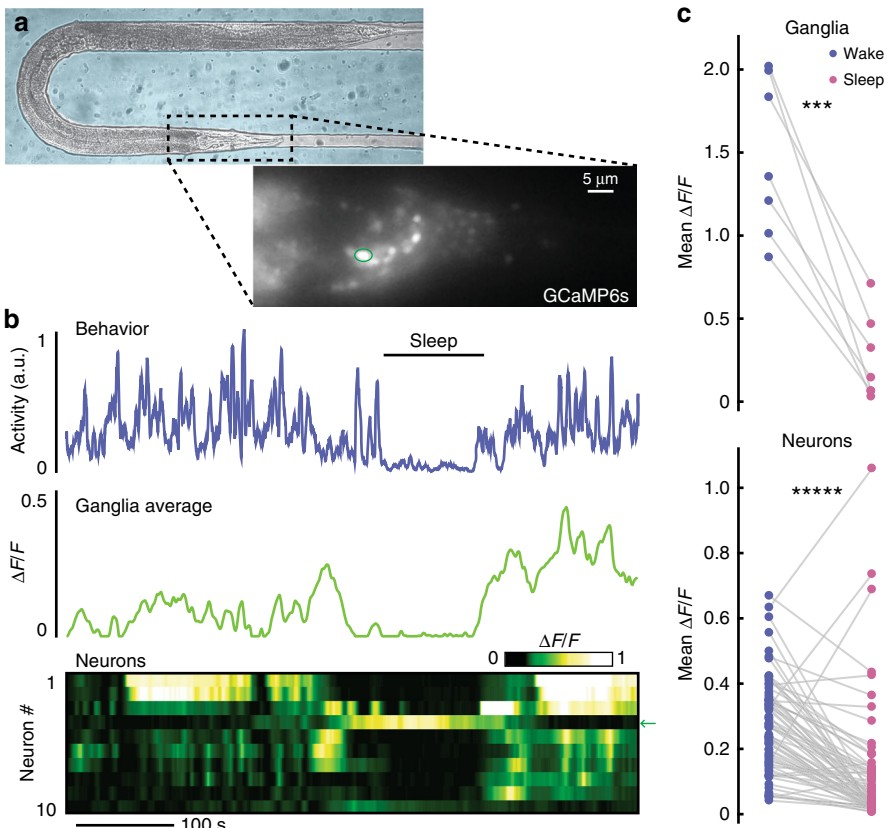

**Fig. 4** A global brain state transition governs *C. elegans* microfluidic-induced sleep. **a** Adult animal immobilized in a microfluidic chamber tailored for whole-brain imaging. (Inset) Single-plane epifluorescence image of an animal with pan-neuronal expression of GCaMP6s. **b** Representative animal shows behavioral quiescence correlates with less neural activity. (Top) Behavioral activity trace quantified by tracking the motion of ten individual neurons (see "Methods" section). (Middle) Average fluorescence across the whole worm head ganglia. (Bottom) Activity of ten individual neurons show a clear brain-state transition and less neural activity during microfluidic-induced sleep. Arrow indicates the circled neuron in **a**, which increases in activity during sleep. **c** Using only behavioral activity, we identified quiescent bouts then quantified neural activity during sleep and wake. During microfluidic-induced sleep, animals exhibited a large-scale downregulation of neural activity across both the entire ganglia and most individual neurons. The neurons chosen for analysis were randomly selected from the field-of-view and are not necessarily the same neurons across every animal. ($n = 7$ animals; ten neurons were tracked per animal; ***$p < 0.001$, *****$p < 0.00001$, paired $t$-test). Source data is available as a Source Data file

large microfluidic device (Fig. 6a). We defined our baseline experimental conditions as: 20 °C cultivation temperature ($T_c$); animals are transferred directly from seeded nematode growth media (NGM) into the microfluidic device; the microfluidic media (M9 buffer) contains no food source; the temperature during imaging is 22 °C.

We first tested how food availability affects the microfluidic-induced sleep phenotype. The presence of food is known to change the swimming-induced quiescence phenotype[43,44]; high quality food induces quiescence[58], and extreme starvation also leads to sleep-wake switching[34,55]. Therefore, we hypothesized that the internal satiety state is a significant regulator of microfluidic-induced sleep. To test this hypothesis, we assayed three satiety states in WT animals: Starved animals were starved for 2 h before being loaded into a chamber with no food; Baseline animals were loaded directly from a NGM food source to a microfluidic device with no food; and +Food animals were loaded from a NGM food source into a microfluidic device with food dissolved in the buffer. As expected, we observed less total microfluidic-induced sleep per worm as satiety increased from Starved (15 ± 2 min, mean ± sem) to Baseline (5.4 ± 0.9 min) to +Food (0.4 ± 0.2 min) (Fig. 6a–b). We also observed that only in the Starved condition did animals show measurably lower behavioral activity during wakefulness (Supplementary Fig. 4A). These results show that satiety is a strong regulator

for microfluidic-induced sleep, which may have evolved to optimize the tradeoff between energy conservation and the search for food.

Other well-known drivers of sleep are noxious environmental stressors, which we also tested in the context of our microfluidic chambers. Heat shock at temperatures >30 °C is a common method to induce cellular stress and *C. elegans* sleep[25,31,35,36,45]. Although our standard microfluidic-induced sleep assays were conducted at only 22 °C, we investigated whether these small changes from $T_c$ could be a possible environmental regulator of quiescence. Indeed, we discovered that increasing the assay temperature to 25 °C, which is still well below commonly used temperatures for heat shock (30–40 °C), increased the total observed sleep 3.5-fold (Fig. 6a, c). Cooling the device to 18 °C (i.e., below $T_c$) had the opposite effect and nearly abolished all microfluidic-induced sleep bouts (Fig. 6c). To show that these changes in phenotype are mediated by *C. elegans* thermosensation, we tested triple knock-out *gcy-23(nj37);gcy-8(oy44);gcy-18 (nj38)* mutants, which show disruption to thermosensation that is specific to the AFD neurons, the primary temperature-sensing neurons in *C. elegans*[59]. In AFD-defective animals we observed no significant change in total microfluidic-induced sleep when raising the temperature from 18 to 22 °C (baseline). Under the same conditions, WT animals showed a 540-fold increase in total microfluidic-induced sleep (Fig. 6c). Furthermore,

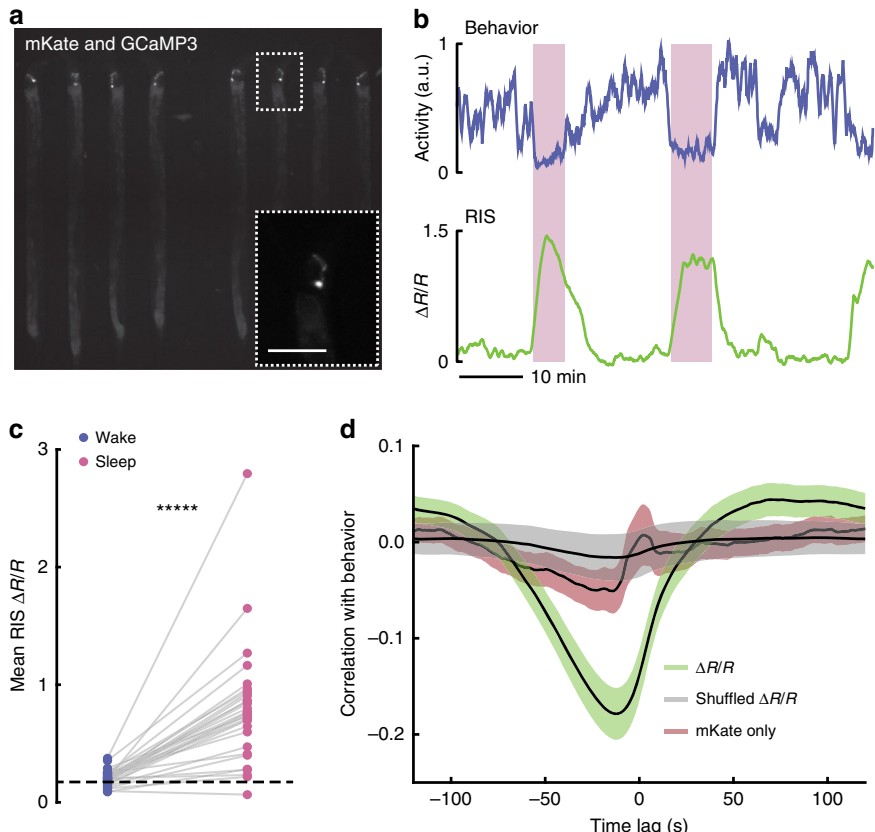

**Fig. 5** The RIS neuron is more active during microfluidic-induced sleep. **a** Fluorescent micrograph of the mKate channel, showing HBR1361 animals immobilized in 50 μm-wide chambers and expressing mKate and GCaMP3 in the RIS neurons. Inset is a zoom-in on a single RIS neuron (scale bar 50 μm). **b** Representative traces of animal behavioral activity (top) and RIS calcium activity (bottom). RIS activity dramatically increases during microfluidic-induced sleep bouts (shaded regions), opposed to the majority of worm brain activity (see Fig. 4). **c** RIS is more active during microfluidic-induced sleep. We automatically detected sleep bouts across animals, calculated mean RIS activity during wake and microfluidic-induced sleep, and quantified mean RIS activity in each behavioral state. Dashed line shows the average RIS activity for animals that did not display a sleep bout. (Data points represent individual animals; $n = 48$ animals total, $n = 31$ animals exhibited at least one sleep bout, $n = 17$ animals did not sleep; *****$p < 0.00001$, paired $t$-test). **d** RIS $\Delta R/R$ activity negatively correlates with animal behavior. This correlation was not seen in shuffled $\Delta R/R$ data or when the mKate channel only was compared with behavior, indicating that RIS correlation with behavior is not a result of movement artifacts. Source data is available as a Source Data file

thermosensory-defective mutants displayed 80% less microfluidic-induced sleep at 22 °C (baseline) and 45% less microfluidic-induced sleep at 25 °C when compared with WT animals (Fig. 6c). These results suggest that AFD neurons transduce changes in environmental temperature and drive downstream sleep-promoting circuits. In addition, temperature had no effect on wake-state behavioral activity for either WT or mutant animals (Supplementary Fig. 4B). These results show that temperature changes significantly less than those typically used for heat shock can dramatically change sleep dynamics and that thermosensory input can act bidirectionally to either promote or suppress *C. elegans* quiescence.

The final environmental factor we investigated was animal confinement. Under baseline conditions, animals swim in large 500 μm-width chambers (Fig. 6d, inset). We found that confining animals to a smaller 110 μm-width chamber, where the primary motion is crawling-like (Fig. 6d inset) leads to the same amount of total microfluidic-induced sleep as the baseline chambers (Fig. 6d). However, when we partially compressed WT animals in small 50 μm-width chambers, we observed nearly a fourfold increase in total sleep compared with baseline (Fig. 6a, d). We hypothesized that this change in phenotype due to immobilization was at least partially mediated by mechanosensory circuits. For example previous reports showed that touch-defective mutants *mec-10(tm1552)* have a decreased ability to sense spatial

patterns in microfluidic arenas[60]. We tested *mec-10(tm1552)* as well and found that when minimal mechanical stress was present (i.e., 500 and 110 μm width chambers), total microfluidic-induced sleep remained insignificantly changed compared with WT (Fig. 6d). However, when *mec-10(tm1552)* animals were compressed, total animal microfluidic-induced sleep decreased by a factor of two compared with WT animals under the same conditions (Fig. 6d), indicating that mechanosensory pathways are necessary for restraint-induced sleep. While animal restraint indeed changed the measured behavioral activity, these differences in sleep between WT and *mec-10* animals cannot be explained by changes in behavioral activity during wakefulness (Supplementary Fig. 4C). In addition, we also tested *mec-4(e1611)* mutants, which are defective in response to gentle touch, but remain sensitive to harsh touch[61]. These animals showed no significant difference in total microfluidic-induced sleep compared with WT when immobilized (Supplementary Fig. 5), indicating that nociceptive mechanosensory pathways are required to drive microfluidic-induced sleep. Together, these results show that *C. elegans* microfluidic immobilization induces sleep behavior through harsh mechanosensory pathways.

Given the dependence of microfluidic-induced sleep on satiety, temperature, and mechanical stress (Fig. 6), we hypothesized that each of these factors could act as stressors that contribute to a slow-onset form of stress-induced sleep. To test this hypothesis,

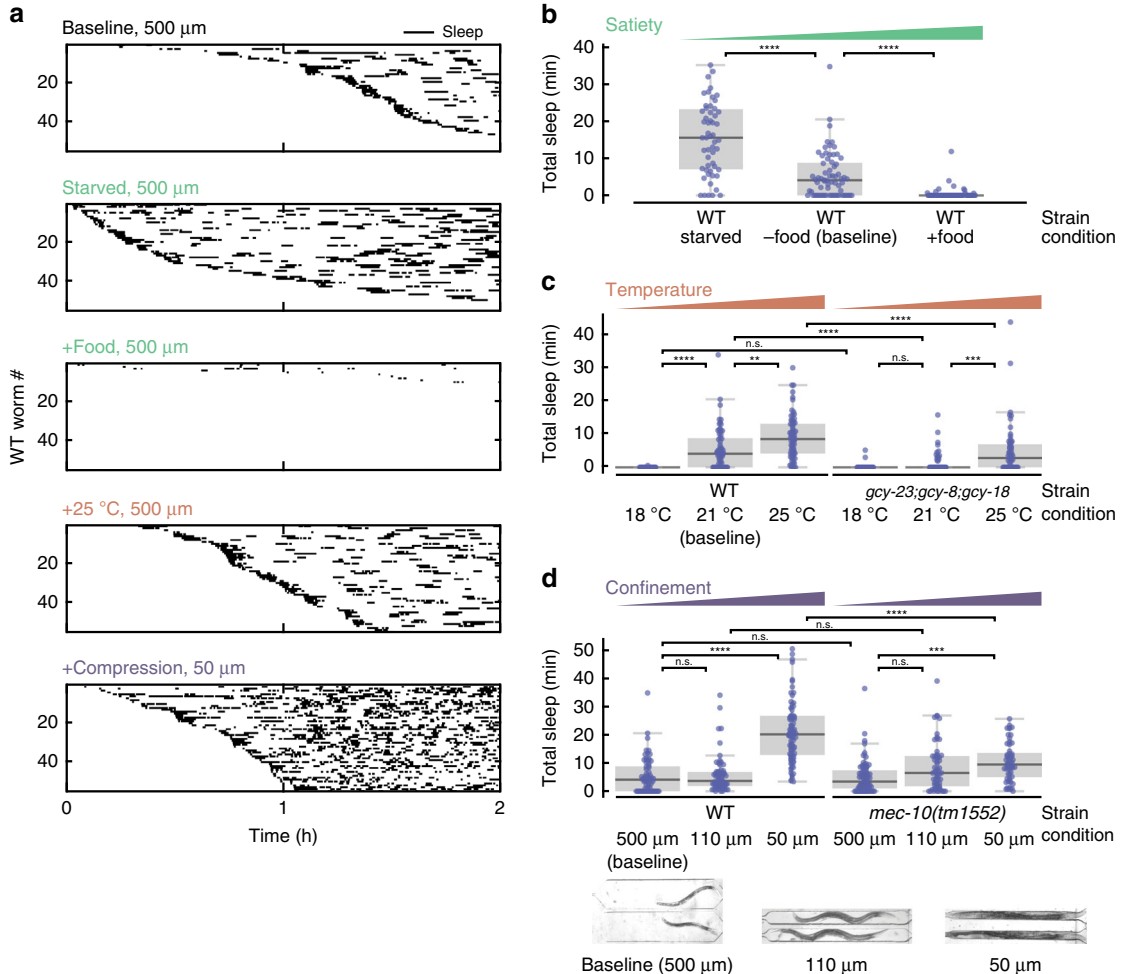

**Fig. 6** Microfluidic-induced sleep is regulated by satiety and multiple sensory circuits. **a** Detected sleep bouts for WT animals in several experimental conditions. Raster plots show detected sleep bouts during a 2 h imaging period. "Baseline" indicates the standard experimental conditions: 500 μm chamber width, no food in the buffer, and a 22 °C temperature. "Starved" indicates animals that were starved prior to the assay. "+Food" indicates conditions in which *E. coli* OP50 was added into the buffer during recordings. "+Heat" indicates imaging conditions where the temperature was raised to 25 °C. "+Compression" indicates that animals were partially immobilized in 50 μm-wide chambers. See micrographs under **d** for chamber geometries. In all cases, the sleep phenotype varies dramatically from the baseline. Only the 55 animals that displayed the most sleep are plotted for clarity. **b** Total WT sleep under varying satiety conditions. As satiety increases from Starved to +Food, animals exhibit less microfluidic-induced sleep (from left to right on the plot the number of animals n = 55, 68, and 67). **c** Total microfluidic-induced sleep under varying temperature conditions. Increasing temperature increases total microfluidic-induced sleep for WT animals. Thermosensory-defective mutants show the same microfluidic-induced sleep phenotype as WT at 18 °C, but significantly less sleep at 22 and 25 °C, indicating that thermosensory input can act to drive or suppress microfluidic-induced sleep (from left to right on the plot the number of animals n = 37, 68, 71, 41, 67, and 60). **d** Total sleep under different confinement conditions. Micrographs show chamber geometries. When WT animals are confined in smaller chambers, they only show an increase in total microfluidic-induced sleep when slightly compressed in 50 μm-wide chambers. *mec-10(tm1552)* mutants show an identical phenotype to WT in 500 μm and 110 μm chambers, but dramatically reduced sleep compared with WT when compressed. These results suggest that nociceptive input to mechanosensory neurons regulates microfluidic-induced sleep (from left to right on the plot the number of animals n = 68, 64, 69, 66, 64, and 57). (ns = not significant, **$p < 0.01$, ***$p < 0.001$, ****$p < 0.0001$; Kruskal–Wallis with a post hoc Dunn–Sidak test). Source data is available as a Source Data file

we assayed DAF-16::GFP animals in the identical environmental conditions shown in Fig. 6. While normally diffuse in the cytoplasm, under unfavorable environmental conditions, DAF-16::GFP localizes to the nucleus[62]. We quantified puncta formation in different environmental conditions and observed that DAF-16::GFP localization correlated with microfluidic-induced sleep phenotypes in only some cases (Supplementary Fig. 6). For example, compared with baseline animals, animals assayed at 25 °C showed greater than a 250% increase in the number of cumulative puncta and animals assayed with food showed almost zero puncta formation (Supplementary Fig. 6), which is in agreement with previous reports of stress responses[62]. These results support the hypothesis that 25 °C and the lack of

food in the microfluidic devices increases animal stress and upregulates microfluidic-induced sleep. However, animals partially immobilized in 50 μm-wide chambers showed no significant difference in the number of DAF-16 puncta compared with Baseline (Supplementary Fig. 6), despite displaying dramatically more total microfluidic-induced sleep (Fig. 6). These results suggest that animal stress may not completely explain the changes in microfluidic-induced sleep we observed during environmental manipulations.

In addition to environmental elements like mechanical compression and temperature that change sleep, we ruled out the possibility that $O_2$ depletion, $CO_2$ buildup, or the buildup of other unknown substances dramatically affect microfluidic-induced sleep

(Supplementary Fig. 7A–C). To rule out these factors we performed two different experiments. In one, we replaced the static buffers (used for all other experiments) with a gentle buffer flow that is expected to stabilize gas concentration levels and remove any animal byproducts. Surprisingly, we observed that the gentle flow led to more sleep (Supplementary Fig. 7A–C). In another experiment, we considered the possibility that the small volume of liquid in different microfluidic chambers could play a role in driving more sleep through the buildup of gasses or other products. To test this hypothesis, we developed a geometry that maintains the same animal restraint but varies the fluidic volume in the worm chambers (Supplementary Fig. 7D). The total sleep in these chambers was not significantly different (Supplementary Fig. 7E–F). These experiments suggest that indeed the mechanical environment is a much stronger driver of sleep than fluidic volume, and not the buildup of metabolic products.

Finally, we also ruled out the possibility that the process of loading animals into the microfluidic device or changing the fluidic environment leads to a stress response that drives stress-induced sleep. We fabricated large microfluidic chambers that are expected to mimic an open volume of liquid (1200 μm long, 500 μm wide, and 90 μm tall), and observed that animals rarely exhibited sleep behavior despite undergoing the same loading process (Supplementary Fig. 7G–I). Therefore, the stress associated with the loading process and change in the fluidic environment alone is not sufficient to drive stress-induced sleep. In fact, reducing this chamber height to 55 μm re-established the sleep behavior, further strengthening our claims that the mechanical environment is a strong driver of microfluidic-induced sleep (Supplementary Fig 7G–I). Overall, our environmental manipulations conclusively show that the animal's satiety state, the mechanical environment, and the local temperature strongly regulate the microfluidic-induced sleep phenotype.

**Whole-brain imaging of transitions in paralyzed animals**. We envision microfluidic-induced sleep as a platform to understand the neural circuits and correlates driving brain-state transitions. To highlight this potential, we performed additional experiments to record from more individual neurons during whole-brain imaging. As shown in previous works, tracking individual neurons in a moving C. elegans brain is a laborious and challenging process[56,63]. Following examples from other groups[23,24,55], we circumvented this challenge by chemically paralyzing animals, performing z-scanning, and capturing the activity of 40–100 neurons simultaneously during volumetric imaging (Supplementary Fig. 8). In a fraction of animals, we observed brain-states that resembled what would be expected from a sleeping animal (decrease in calcium activity in all but a few neurons, see Fig. 4). However, in this paralyzed preparation we cannot unequivocally claim that these states correspond to sleep because we cannot correlate the calcium imaging with animal behavior. Nevertheless, our data is consistent with our expectations for how paralysis should affect the sleep state given the role of mechanoreception. We found that the probability of observing a putative sleep state during a 10-min experiment dropped from a rate of ~25% in behaving animals to <5% in paralyzed animals. This result is consistent with our conclusion that mechanosensation of the environment facilitated by worm movement is a primary driver of microfluidic-induced sleep. Though we are limited here by a relatively slow imaging rate (<3 vol/s) and poor z-sectioning with epifluorescence imaging, these experiments show a path towards using rapid volumetric imaging techniques[64] to not only capture brain-wide activity but also enable single-neuron mapping to the C. elegans connectome.

## Discussion

Here, we described a spontaneous C. elegans brain and behavioral state transition that is unique to microfluidic chambers, which facilitates whole-brain imaging and precise regulation of the environment. Microfluidic-induced sleep regulation appears to be at least partially linked to a slow-onset form of stress-induced sleep (Supplementary Fig. 6), and may also be related to swimming-induced quiescence[40,42,43]. Despite these similarities, the behavioral dynamics of quiescence that we measured is quantitatively distinct from these previously reported worm quiescent behaviors (Fig. 1), although it is possible that changes in sleep onset time and bout duration in microfluidics are affected by collisions with the chamber walls or other physical cues. We showed that during microfluidic-induced sleep animals exhibit many behavioral properties of sleep, including a stereotyped posture (Fig. 2a), reversibility (Fig. 2b), a decreased response to sensory stimuli (Fig. 2c–d), and homeostatic rebound (Supplementary Fig. 2). This quiescence behavior is also dependent on known C. elegans sleep-promoting neurons, ALA and RIS (Fig. 3). In addition, we found that thermosensory input via the AFD neuron acts as a bidirectional controller of sleep; cooler temperatures promote wake, while warmer temperatures promote sleep (Fig. 6). Furthermore, we showed that animal restraint can act through mechanosensory pathways to drive sleep behavior (Fig. 6). Finally, a global downregulation of neural activity, with the exception of a few neurons, underlies microfluidic-induced sleep behavioral transitions (Figs. 4 and 5, Supplementary Fig. 8).

Importantly, our results show the drastic effects of microfluidic environments on C. elegans behavior. Microfluidic assays are commonly used to study nematode biology[65]. Sleep in C. elegans is also commonly studied by confining animals to microfluidic chambers[43,44,66]. Our data show that the conditions of these micro-environments can strongly affect the sleep phenotype and thus the microfluidic chambers should be carefully designed and their effects should be considered when interpreting the results of these experiments. However, although we observe changes in sleep behavior in the context of microfluidics—and most notably during increased confinement—quiescence behavior may increase in standard agar environments as well due to changes in animal confinement, such as agarose-based confinement chambers[66].

Furthermore, it is not difficult to imagine that C. elegans has evolved to use environmental cues to regulate quiescence behavior. One unifying hypothesis for C. elegans microfluidic-induced sleep regulation could be rooted in energy conservation. Factors that decreased microfluidic-induced sleep, such as food and cooler temperatures, may indicate favorable conditions for active roaming behavior. Conversely, factors that increased sleep, such as elevated temperatures and animal restraint, may indicate more harmful environments in which roaming and expending energy is not an optimal survival strategy. Furthermore, the changes we made to the environment may change the animal's metabolic state. C. elegans metabolism is known to increase with increasing temperature[67]. Therefore, it is possible that all changes in the sleep phenotypes we observed were rooted in a tradeoff between expending energy to move and the expected value of finding food.

Many questions remain with respect to C. elegans sleep and the role of sensory inputs in driving sleep behavior. The interaction between multiple sensory inputs, the ALA/RIS neurons, and diverse behavioral outputs are likely highly non-linear systems involving multiple signaling pathways and neuromodulators[25,29,34–36,44]. Future work will help answer these questions and determine the relationship between ALA and RIS and whether these neurons are critical for driving sleep in all the environmental conditions tested in this report (Fig. 6). Furthermore, ALA and RIS themselves are known to respond to mechanical stimulation[66,68], potentially providing a mechanism for why animal confinement drives sleep

behavior. These sleep-driving neurons that also act as sensory neurons are a prime example of the multimodal nature of the *C. elegans* nervous system, which adds another element of complexity when dissecting the sensory and sleep circuits involved in environmentally driven sleep.

In a broader context, we have identified microfluidic-induced sleep as a behavior in which experimentally controlled external inputs strongly affect how frequently an animal transitions between behavioral states. Therefore, this sleep behavior provides a testbed for interrogating the biological principles that regulate brain and behavioral state transitions. These studies are made possible by the combination of whole-brain imaging and behavioral recordings with simultaneous control of the microfluidic environment. Combining these modalities in a high-throughput pipeline will lead to a large library of simultaneous behavioral and brain-wide data under many environmental and physiological conditions. These data, combined with computational techniques, would provide a rich resource to understand how many circuits across the *C. elegans* nervous system regulate behavioral states.

## Methods

**C. elegans strains and maintenance**. All *C. elegans* strains were maintained at 20 °C on standard NGM seeded with *E. coli* OP50 as the food source. All experiments were performed with day 1 adult animals at 21–22 °C unless otherwise stated. The strains used for each experiment are as follows:

Figure 1: N2

Figure 2: N2

Figure 3: N2, IB16 *ceh-17(np1)* I; HBR227 *aptf-1(gk794)* II

Figure 4: AML32 (*wtfIs5 [prab-3::NLS::GCaMP6s; prab-3::NLS::tagRFP]*)

Figure 5: HRB1361 (*goeIs304[pflp-11::SL1-GCaMP3.35-SL2::mKate2-unc-54-3′UTR, unc-119(+)]*)

Figure 6: N2, IK597 *gcy-23(nj37);gcy-8(oy44);gcy-18(nj38)* IV, ZB2551 *mec-10 (tm1552)*

Supplementary Figure 1. N2

Supplementary Figure 2: N2, HBR1374 *goeIs307[pflp-11::ArchT::SL2mKate2-unc-54- 30UTR,unc-119(+)]*; *goeIs304[pflp-11::SL1-GCaMP3.35- SL2::mKate2-unc-54-30UTR, unc-119(+)]*

Supplementary Figure 3: N2, IB16 *ceh-17(np1)* I; HBR227 *aptf-1(gk794)* II

Supplementary Figure 5: N2, CB1611 *mec-4(e1611)*

Supplementary Figure 6: TJ356 (*zIs356 [daf-16p::daf-16a/b::GFP + rol-6 (su1006)]*)

Supplementary Figure 7: N2

Supplementary Figure 8: AML32 (*wtfIs5 [prab-3::NLS::GCaMP6s; prab-3::NLS:: tagRFP]*)

**Microfluidic device fabrication**. Standard photo- and soft-lithography techniques were used to fabricate microfluidic devices[65]. Microfluidic geometries were custom designed in CAD software. Most photomasks were transparencies (CAD/Art Services Inc.), but glass photomasks (Front Range Photomask) were used for higher-resolution devices (Supplementary Fig. 7). All master molds were fabricated using SU-8 2075 (MicroChem). The SU-8 height for worm behavioral channels was 75 μm (spin: 20 s–500 rpm, 30 s–3000 rpm), but we used a height of 50 μm (spin: 20 s–500 rpm, 30 s–4000 rpm) for whole-brain imaging devices to further constrict animal movement. We used polydimethylsiloxane (PDMS) Sylgard for all microfluidic chips. All behavioral chips were double-layer devices to incorporate push-down valves for sealing the chamber entrances or delivering mechanical stimuli. The bottom worm layer (20:1 ratio, spin: 930 rpm for 30 s) was bonded to the upper valve layer (10:1 ratio) using a 30 s exposure to oxygen plasma (200 W, 330 mTorr), then baked together for at least 12 h. The PDMS devices were permanently bonded to either a standard glass slide for behavior, or a 300-μm-thick quartz wafer (NOVA Electronics Materials) for whole-brain imaging.

**Behavioral quantification and sleep detection**. To quantify *C. elegans* activity, we used frame-by-frame subtraction using custom MATLAB (MathWorks) scripts. Not only is frame-by-frame subtraction less computationally expensive than other metrics such as posture or nose-tracking, but we found overall that this technique is less prone to the noise associated with small errors in tracking specific *C. elegans* body points. This method subtracts consecutive frames from one another, then counts the number of pixels that substantially change value. Worm movement leads to a large number of pixels that change from frame-to-frame. Quiescent animals, which move very little, lead to few pixels that change values from frame-to-frame. We performed frame-by-frame subtraction, drew regions-of-interest around each animal, then counted the number of pixels in the ROI that changed by a value >30 (a number well above the noise level of the CMOS sensor). This yields a raw activity trace for each animal. We also used standardized methods to normalize

activity traces[43,44], which accounts for changes in brightness across the field-of-view, and allowed us to set a consistent sleep detection threshold across populations. In most cases, we smoothed activity traces across 20 s, then normalized to the top 95th percentile, yielding a normalized activity trace for each animal with values approximately between 0 and 1. For analysis that required finer timescales on the order of <20 s (Fig. 2a–c), we did not smooth activity traces (see Reversibility and Decreased Response to Stimuli subsection).

Once we normalized the activity trace, we thresholded the data to detect sleep. The threshold depended on the microfluidic geometry. For example, in large microfluidic chambers (Supplementary Movie 1), movement could still be detected during microfluidic-induced sleep as animals drifted across the chamber. In smaller microfluidic geometries (Supplementary Movie 2), worm activity was already significantly constrained, so a stricter threshold was needed to detect sleep. We determined thresholds by manually scoring 20 sleep bouts, calculating the mean activity during those bouts, then doubling the mean activity. The thresholds used for each geometry were: WormMotel: 0.15, 500 μm microfluidic chambers: 0.15, 110 μm microfluidic chambers: 0.08, 50 μm chambers: 0.06. A minimum sleep bout time of 30 s was used to reduce false detections. In addition, during a sleep state brief animal twitching lasting <15 s was not counted as a wake period.

**Standard microfluidic behavioral assays**. Unless otherwise stated, behavioral assays took place on an enclosed AmScope SM-2T-LED stereo microscope. Red transparency was used to filter out LED wavelengths that potentially affect animal behavior. Imaging was performed at 3 fps with either a Point Grey Grasshopper (GS3-U3-23S6M-C) or Basler Ace (acA1920-40um) CMOS cameras, which have nearly identical sensor specifications. While the experiments were typically carried out in a room held at 20 °C, the heat from the LED raised the microfluidic device temperature to 21–22 °C (the Baseline temperature in Fig. 6).

M9 buffer was used for all experiments, with no food added unless otherwise stated. During standard experimental conditions, we used a hair pick to transfer day 1 adults directly from seeded NGM into the buffer of an open syringe cap. We then suctioned animals into Tygon tubing that led to the microfluidic chambers. The process of loading 6–12 animals into the chambers (depending on the geometry used) typically took <5 min. A push-down valve closed off the entrance of all worm chambers to prevent animals from escaping during imaging. After use each day, devices were flushed with ~6 mL of DI water, sonicated for 10 min, flushed again with DI water, boiled for 15 min, and flushed a final time before storage at ~80 °C overnight.

**WormMotel assays**. A 48-well WormMotel molded from PDMS was provided by the Fang-Yen Lab[47]. Only 12 wells were used simultaneously. Prior to use, the PDMS was exposed to oxygen plasma for 30 s to make the surface hydrophilic. Each well was then filled with 8 μL of M9 buffer. Using a hair pick, we removed individual animals from seeded NGM, washed them in an M9 droplet, then transferred them into the WormMotel wells. For imaging, the PDMS was inverted and reversibly sealed on a glass slide during imaging (see Churgin et al. [44,47]). The glass slide was treated with Rain-X to prevent condensation. The device was illuminated obliquely with three AmScope Goose-neck LEDs that were filtered with red transparency. As with microfluidic devices, the temperature during imaging was 21–22 ºC. We imaged from below at 3 fps with a Point Grey Grasshopper (GS3-U3–23S6M-C).

**Heat shock**. Device setup and animal loading was performed as previously stated (see Standard Microfluidic Behavioral Assays subsection). However, the device was preheated to 30 °C with two Peltier heaters placed on each side of the glass slide. Heat was applied for the initial 30 min of imaging, after which the current through the Peltier heaters was reversed such that the device was rapidly brought to the standard 22 °C. This heat shock protocol was adapted from that used for agar plates in Hill et al. [37].

**Posture and body curvature analysis**. Similar to previous reports[48], Custom MATLAB scripts were written to quantify body curvature in the Large microfluidic data set from Fig. 1 (where animals are confined to microfluidics, but have enough room to swim). We limited our analyses to the first 2 h of imaging, when a minimal number of eggs are present. Many eggs in the chamber can make it computationally difficult to distinguish between eggs and the animal body. For every video frame, we subtracted the background (an image of the chambers with no animals present) and increased the frame contrast (MATLAB imadjust). These pre-processing steps yielded a high contrast between the animal bodies and image background. We then binarized the video (MATLAB im2bw), removed small objects (MATLAB bwareaopen), smoothed the binarized frame (MATLAB imgaussfilt), skeletonized the smoothed frame (MATLAB bwmorph), and removed spurious pixels (MATLAB bwmorph). These steps gave skeletonized versions of the worm bodies. We used these skeletons to calculate the average body length and placed 20 equally spaced points along the centerline. Occasionally (<5% of frames), this centerline fit performed poorly (e.g., animals completely curl on themselves), but this could be detected by calculating the body length in every frame. Frames where the body length was <80% of the average were not used to calculate curvature. From the centerline points, we fit a circle to every three adjacent points,

yielding 18 curvature values along the body (the inverse of the circle radii). These 18 curvature values were averaged and normalized to the body length to give a dimensionless value for the mean animal body curvature in every frame. For every animal, we then smoothed the curvature data over 20 s and found the average curvature during sleep and wakefulness.

**Reversibility and decreased response to stimuli**. Reversibility assays were conducted on an inverted Nikon microscope and performed with 50 μm-wide microfluidic chambers (Supplementary Movie 3). Animals were loaded into the chambers and left for between 30 and 60 min. Upon visual confirmation of sleep, we initiated a protocol of 60 s of imaging at 10 fps. At 30 s, a strong light stimulus (460 nm light at 5 mW/mm2) was presented for 5 s to awaken animals. Only animals that were asleep for the full 30 s prior to the light stimulus were kept for analysis.

To test for a decreased response to stimuli, we fabricated 110 × 1100 um microfluidic chambers to confine animals (Supplementary Movie 4, 5). In addition to the push-down valve used to keep animals in each chamber, we incorporated two push-down valves used for mechanical stimulation. The two valves ensured that animals were almost always in contact with at least one valve during pressurization. For strong and weak stimulation, we used a valve pressure of 30 and 15 psi, respectively. Seven groups of animals were used for weak stimulation and four for strong stimulation; each group consisted of 8–12 animals. During the hour-long assay, animals received a 0.5 s valve pulse every 3 min. During postprocessing, we analyzed animal activity around each stimulation timepoint. For the 10 s prior to each stimulation timepoint, we classified animals as quiescent, slow moving, or fast moving. Two categories for wake behavior (i.e., slow moving and fast moving) were necessary because a behavioral response could not be detected in animals that were already moving with a high activity. That is, an animal that was clearly awake and moving with high activity in the microfluidic chamber did not display increased activity after receiving mechanical stimuli. However, a detectable behavioral response was apparent in wake, but low-activity animals. We used an activity threshold of <0.08 for quiescent animals, 0.08–0.35 for low-activity animals, and >0.35 for high-activity animals (the average animal activity across all traces was 0.36). This classification was performed for each stimulation (20 stimulations for each animal). These classified groups comprise the heatmaps showed in Fig. 2b. Note that because animals are more likely to be in the wake state, this classification method leads to lower n-values for the sleep condition. From each animal in each group, we also calculated the mean activity for 10 s post stimulation, excluding a 2 s period where the microfluidic valves caused movement artifacts, which make up the data shown in Fig. 2c.

**Homeostasis and RIS optogenetic inhibition**. This protocol was adapted from Wu et al. [34]. HBR1374 animals expressed the light-gated proton pump ArchT in the RIS neuron and were raised on NGM seeded with 0.2 mM all-trans retinal (+ATR). Control animals were also the HBR1374 strain but were not raised in ATR (-ATR). Experiments were performed on an inverted Nikon microscope. Animals were loaded in 110 μm-wide chambers (see Fig. 6 inset) and imaged at 4 fps with a Basler Ace (acA1920–40um) CMOS camera. Experiments lasted 2 h. Between 1 and 1.5 h, we illuminated animals with 565 nm light (3 mW/mm2).

**Whole-brain imaging in behaving animals**. Microfluidic devices were fabricated as previously described (see Microfluidic Device Fabrication subsection), and geometries were modeled off of previous methods for whole-brain imaging[23,24,55]; however, animals were not chemically paralyzed. We used transgenic animals with pan-neuronal expression of GCaMP6s in the nuclei of all neurons[56,63]. Experiments were performed on an inverted Nikon microscope with a ×40 water-immersion objective (NA = 1.15). An Andor Zyla 4.2 USB 3.0 sCMOS captured the images at 5 fps (50 ms exposures, 2 × 2 binning). Because blue light can wake C. elegans, when animals were immobilized in the chambers, we first initialized a habituation protocol. Here, we flashed the 460 nm excitation light for 0.15 s at 1 Hz for 30 min. This habituated animals without photobleaching the calcium indicator. After the habituation period, we imaged continuously for 15 min, but only analyzed the first 10 min of data due to photobleaching. With this protocol, we captured a sleep-wake transition in ~25% of animals.

To analyze data, we chose ten neurons at random to track by hand in FIJI. An 8 × 8 pixel ROI was then used for each neuron in each frame. A neuron near the center of the ganglia was used to draw an ROI around the head ganglia to calculate average brain fluorescence in each frame. We calculated animal activity in two ways: by frame-by-frame subtraction (sampling the frames a 3 Hz), and by calculating the average displacement of the tracked neurons. Averaged and normalized neuron displacement make up the behavioral traces in Fig. 4, Supplementary Movie 6. Both methods were prone to fluctuating baselines that were not present during behavioral-only recordings, making it difficult to set a standard threshold across animals. However, clear quiescent periods were subjectively apparent. Therefore, for each animal we plotted only the animal behavioral activity and selected sleep bouts manually. As before, only bouts longer than 30 s were counted. We then calculated the mean ganglia and neuron fluorescence during sleep and wake for each worm.

Behavioral activity was smoothed over 3 s and normalized as previously described (see Behavioral Quantification and Sleep Detection subsection). Raw

ganglia and neuron activity was smoothed over 3 s before calculating $\Delta F/F$. We denote $\Delta F/F$ as $[F(t) - F_0(t)]/F_0(t)$ where $F_0(t)$ is the minimum fluorescence value up to time $t$.

**Whole-brain imaging in paralyzed animals**. Whole-brain imaging in paralyzed animals closely followed the protocol for behaving animals (see Whole-Brain Imaging in Behaving Animals subsection); however, 5 mM tetramisole was added to the buffer to chemically paralyze animals, similar to previously reported methods[23,24,55]. In addition, the ×40 objective was mounted onto a piezo scanner (Physik Instrumente). We used MATLAB to control a data-acquisition box (National Instruments) that output voltage waveforms to trigger both the piezo scanner and Zyla camera. The DAQ output a sawtooth waveform to the piezo, which enabled scanning across 30 μm in the z-axis. At 12 different points along the scan, the piezo would briefly settle for 30 ms while the DAQ also triggered a 10 ms camera frame acquisition. This approach led to a volumetric imaging rate of 2.84 vol/s over 10 min periods.

Following imaging, we used a custom MATLAB GUI to manually choose neuron locations. The GUI allowed users to look at max projections, volumes, or single imaging planes across the duration of the recording. Consecutive planes or volumes could also be averaged to reduce noise. This allowed for the x-y-z positions of neurons to be manually located and stored. The average of a $3 \times 3 \times 3$ pixel ROI was used around each neuron when extracting mean fluorescence. Raw neuron activity was smoothed over 3 s before calculating $\Delta F/F$. We denote $\Delta F/F$ as $[F(t) - F_0(t)]/F_0(t)$ where $F_0(t)$ is the minimum fluorescence value up to time $t$.

**RIS Imaging**. Overall, 7–10 animals per trial (six trials total) were confined and imaged simultaneously under ×10 magnification in the 50-μm-wide chamber geometry. HRB1361 animals expressed both GCaMP3 and mKate in RIS neurons under the FLP-11 promoter. Experiments took place on an inverted Nikon microscope, with dual excitation and emission for simultaneous two-color imaging (FF01–468/553–25 and F01–512/630–25 BrightLine dual-band bandpass filters for excitation and emission, respectively). An X-Cite XLED1 light source provided both 460 and 565 nm excitation light and a Tucam image splitter (Andor) split the mKate and GCaMP channels onto two Andor Zyla 4.2 CMOS cameras. We imaged animals for 2 h and used a data-acquisition box (National Instruments) controlled with custom MATLAB scripts to simultaneously trigger 10 ms camera exposures and 30 ms XLED flashes at 0.5 Hz (3 × 3 binning). Following experiments, we flushed animals from the device and recorded the background for each color channel.

For postprocessing, we used the mKate channel to quantify behavior and track the location of RIS neurons. After subtracting background, we calculated animal activity by drawing an ROI around each animal and performing the same frame-by-frame subtraction method as previously described (see Behavioral Quantification and Sleep Detection subsection), but here we used a pixel change with a threshold of 400 because of the high dynamic range of the Zyla sensor. After normalizing behavioral activity, a threshold of 0.2 was used to detect sleep bouts, with a minimum sleep time of 30 s. To detect the RIS location in each frame, we again subtracted the image background. We then isolated and binarized each animal ROI, leaving only the RIS neuron and, occasionally, pieces of its processes. The largest object in the ROI was the RIS soma, which we used to attain the soma centroid. To get the average RIS fluorescence in the frame, we drew a $25 \times 25$ pixel ROI around the centroid in each channel and averaged the 20 largest pixel values. We normalized raw fluorescence in each channel by calculating $\Delta F/F = [F(t) - F_0(t)]/F_0(t)$, where $F_0(t)$ is the minimum fluorescence value up to time $t$. Following this normalization, we calculated the ratio of each color channel $R = (\Delta F/F)_{GCaMP}/(\Delta F/F)_{mKate}$. Finally, the reported ratio values are $\Delta R/R = [R(t) - R_o]/R_o$, where $R_o$ is the lower 20th percentile value.

Using behavioral data (from frame-by-frame subtraction), we detected sleep bouts and calculated the average RIS $\Delta R/R$ activity during wake and sleep (Fig. 5c). To calculate correlation values, across all animals we split behavioral and fluorescence data into 10 min windows and calculated the fluorescence correlation with behavior in each window. $\Delta R/R$ data showed a strong negative correlation with behavior (Fig. 5d, green trace). Randomly shuffling the $\Delta R/R$ windows as a control removes this strong correlation (Fig. 5d, gray trace). To test for movement artifacts that could possibly lead to a correlation between fluorescence and behavior, we performed the same unshuffled analysis with the mKate channel only (Fig. 5d, red trace).

**Environmental control**. The microfluidic setup during environmental manipulation was identical to the previously stated (see Standard Microfluidic Behavioral Assays subsection); however, for each condition we manipulated individual aspects of the environment. The baseline phenotype in 500-μm-wide chambers (Supplementary Movie 1, Fig. 6) was chosen because animals could move freely in a manner similar to swimming, thereby reducing the overall effect of the microfluidic confinement.

Most assays involved transferring animals directly from seeded NGM into the microfluidic device. However, under Starved conditions we used a hair pick to remove animals from seeded NGM, washed them in a droplet of M9 buffer, then placed them onto a fresh, unseeded NGM plate. After 2 h, animals were loaded into the microfluidic device with no food in the buffer. Therefore, animals were deprived of a food source for 2 h when imaging began and did not have access to food during the experiment.

Most assays were also performed with no food in the M9 buffer. However, under +Food conditions, we dissolved 4.56 mg/mL of freeze-dried OP50 (LabTie) into the M9 buffer, providing a food source for animals throughout the imaging time.

In +Compression conditions, animals were confined to 50 × 1100-um-wide microfluidic chambers (Supplementary Movie 2).

As previously stated, the standard imaging temperature due to LED lighting was 21–22 °C (see Standard Microfluidic Behavioral Assays subsection). We controlled temperature in the same manner as heat shock experiments, using Peltier devices to heat or cool the microfluidic chips to 24–25 or 17–18 °C, respectively.

**DAF-16::GFP Imaging**. Experiments took place over the course of 2 h on Nikon inverted scope with an Andor Zyla 4.2 USB 3.0 sCMOS and ×4 magnification for multi-worm imaging. Environmental control took place as described in the Environmental Control subsection. Every 5 min, we captured seven frames (20 ms exposures) at 1 Hz with 1 × 1 binning. Otherwise, animals were in darkness. Following imaging, we flushed animals from the chamber and recorded background frames.

Puncta quantification took place similar to previous methods with custom MATLAB scripts[69]. We subtracted the background from each frame and smoothed the image with a median filter. We thresholded this image to detect animal bodies and drew an ROI around each. The smoothed image was further processed with a 3 × 3 high frequency filter, which enhanced the contrast of puncta and allowed us to binarize the image to detect most DAF-16 aggregations in the frame. For each animal ROI, we then found puncta by counting all objects with a size between 2 and 30 pixels. To mitigate the effects of animal movement, which caused some animals to appear blurred and reduced puncta count, of the seven frames that were captured every 5 min, we averaged the three largest puncta counts for each ROI. This average makes up the final puncta count for each animal at the given timepoint. The puncta count reported (Supplementary Fig. 6) are likely lower than the true count that would be attained with high-magnification imaging. Low magnification not only increases the number of animals imaged per trial but was also necessary to capture the full length of the large microfluidic chambers. Because we applied the same algorithm to all environmental conditions, we expect the reported trends to remain true in high-magnification conditions.

**Quantification and statistical analysis**. Boxplots with scatterplots overlaid were constructed in MATLAB. For the boxplots, the top and bottom edges of the box indicated the 25th and 75th percentiles, respectively; the whiskers extend to the most extreme data points not calculated as outliers; and the gray line in the box indicates the median.

All statistical tests are indicated in figure legends. Almost all statistical comparisons were between more than two groups with unequal variances. Thus, the non-parametric Kruskal–Wallis test with a post hoc Dunn–Sidak test was used to test for statistical significance. For data in which we tested whether one group displayed phenotypic differences (either behavioral or neural) during wake and sleep, we used the paired *t*-test.

**Reporting summary**. Further information on research design is available in the Nature Research Reporting Summary linked to this article.

## Data availability
All raw data (e.g., raw animal activities from frame-by-frame subtraction) will be made publicly available at https://rice.box.com/v/Celegans-Microfluidic-Sleep. Raw video files will be provided upon request. Source data underlying Figs. 1–6 and Supplementary Figs. S1–S8 are available as a Source Data file.

## Code availability
Custom MATLAB scripts used for data analysis will be provided upon request.

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

## Acknowledgements

We thank the Fang-Yen lab for providing the WorMotel device. Henrik Bringmann provided the HRB1361 strain for RIS imaging and HBR1374 strain for optogenetics. We thank Anne Hart, David Raizen, and Henrik Bringmann for helpful discussions. Several strains were provided by the CGC, which is funded by NIH Office of Research Infra-structure Programs (P40 OD010440). D.L.G. is funded by the National Science Foun-dation (NSF) Graduate Research Fellowship Program 1842494 and a training fellowship from the Keck Center of the Gulf Coast Consortia on the NSF IGERT: Neuroengineering from Cells to Systems 1250104. D.L.G. and J.Z. are funded by the Smalley-Curl Institute's Student Training for Advising Research program. We also thank the Rice Shared Equipment Authority where devices were fabricated.

## Author contributions

D.L.G. conceived, performed, and analyzed experiments. J.Z. performed whole-brain imaging experiments. B.F. performed experiments. J.T.R. directed the research. D.L.G. and J.T.R. co-wrote the paper. All authors read and commented on the manuscript.

## Competing interests

The authors declare no competing interests.
