## [Peer Review File · Nature Communications]

Reviewers' comments:

Reviewer #1 (Remarks to the Author):

In this manuscript Gonzales and colleagues report so far undescribed experimental conditions that promote sleep in adult *C. elegans*. The best described sleep state in *C. elegans* is developmentally timed sleep (DTS), but during recent years multiple labs have established additional sleep paradigms, e.g. stress and starvation induced sleep. The authors show here that constraint in microfluidic devices promotes spontaneous bouts of immobility, termed here mu-sleep. They validate that mu-sleep fulfills behavioral criteria of sleep (reversibility, increase in arousal threshold, homeostasis). Imaging neuronal activity via a Ca⁺⁺-indicator, either pan-neuronally expressed or specifically in the sleep promoting neuron RIS they show that mu-sleep likely exhibits the same neuronal activity signature that was previously reported for DTS. Using genetics, the authors show that mu-sleep requires genetic pathways that were previously shown to be required for DTS and stress induced sleep. Finally, some evidence is provided for the implication of mechano-receptors in the induction of mu-sleep. Most experiments are sophisticated and statistical analyses are sound. One exception are the brain wide imaging experiments, previous work recorded brain wide activity during sleep at single cell resolution, while in the current study wide field epi-fluorescence imaging can only provide a crude estimate about the neuronal activity changes upon mu-sleep onset. However, for the point that the authors want to make here, this seems sufficient to me.

The paper provides an incremental step towards understanding regulation of sleep in *C. elegans*. Whether mu-sleep is a distinct new sleep state or a mild form of stress induced sleep remains unclear. Moreover, the factors that trigger mu-sleep remain enigmatic. The various constraining conditions used here vary greatly in how they could affect the animals; also, the ethological relevance of mu-sleep is unclear and only vaguely discussed by the authors. The paper is of special interest to the community of *C. elegans* researchers who uses micro-fluidic devices for neuronal imaging under even more constraint conditions; the prevalence of a sleep state under these conditions is important for the interpretations of their results. This aspect should be definitely highlighted in the discussion.

Specific comments:

I think it is likely that flushing worms into the microfluidic devices could cause some mechanical damage. Is mu-sleep simply stress induced sleep in response to this possible damage? If yes, this would change the major conclusions of this paper. To control for this, the authors could flush the worms out of the device and then measure quiescence under unconstrained conditions.

Minor comments:

Results section: "is typically defined (...)": this sentence seems to suggest that there is a specific reason to have different criteria based on whether it is adult sleep, which I don't think it's justified.

Fig.1:

I have big doubts about the very first sentence of the legend/title: Basically, would like to know whether the onset of sleep maybe simply correlates with the volume of the different chamber/well (by testing a couple more chambers with intermediate and/or larger volumes), and/ or the number of times the animal "touches" every wall, before falling into immobility. Very interestingly, it seems that one possible explanation for small chamber, large chamber and worMotel-phenotypes is simply a scaling of the very same reaction: the worms try for some time to move (sampling the liquid environment around it and/or the walls), until it realizes it is in an unescapable environment, with no interesting stimuli around. In such environments, trying to move and expending energy is pointless. The realization takes longer if the environment is larger.

Could the duration of the Q bouts be affected by how many times the worm bump or is drafted

into the walls (leading to a stimulation that interrupts sleep, waking up the worm)? If this is the case, this would be the simple reason why the Q bouts are longer in the worm. It seems like only very large body movements would be caught while imaging in the smallest chamber, of 50µm (there is a video where the worm slightly moving its tail is labelled as sleeping). Might this be the case? So: How would the Q bouts, labelled based on the high RIS activity, compare with the behavioral estimate based on frame subtraction?
Legend: HS is defined as "first hour of heat shock" but is actually half a hour.

Fig. 2:
Why so different n numbers for the 2C?

Fig. 4:
Not clear whether the neurons quantified for their activity in C are the same random 10 neurons picked for tracking and for detecting movement (see Methods).

Fig. 5:
Fig.5A: I suggest to make an inset zooming in on the heads
Fig.5B: Shaded regions are invisible upon print

Fig. 6:
Legend: "no food IN the buffer" ("in" is missing)

Fig. Suppl. 3: Panel labelling goes A-C-D

Methods:
Why for RIS imaging the threshold for sleep bout detection is 60s? It was 30s for both behavior and whole-brain imaging.
DAF-16(...) referring to Suppl. Fig1 wrongly (it's Suppl. Fig.3)

Reviewer #2 (Remarks to the Author):

In this manuscript by Gonzales and colleagues, the authors describe a new platform for studying sleep in *C. elegans*. They demonstrate that μ Sleep is phenotypically and mechanistically similar to SIS (and DTS), and the microfluidic approach provides opportunities for studying sleep mechanisms both with regard to behavioral detail and imaging. They go on to show that μ Sleep is modulated by diverse sensory inputs, including temperature and mechanical stress. The work is interesting and describes an important new platform, though does not reveal much in the way of new sleep biology.

I have some major concerns and suggestions regarding the experiments.

1. I am not convinced with how the authors define what they count as sleep (quiescence >30 seconds). This is a critical issue and should be discussed prominently in the results, with evidence for how this was selected. For example, if the authors look at ALL quiescence, regardless of length of time, do trends look the same but the data is just noisier (the authors say in the methods "A minimum sleep bout time of 30 s was used to reduce false detections"). What if counting 5s or 15s bouts as sleep altered the conclusions of some experiments? We would then need experiments to justify 30s as the beginning of biological relevance for sleep. Again, if included shorter bouts simply increases noise without altering the overall patterns observed in a few of the experimental paradigms, I would be more convinced.

2. The authors need to test sleep deprivation directly. If the animals do not exhibit homeostatic

rebound, I do not believe it would invalidate the model. But in the current version, they say only that sleep deprivation can produce a stress response so they don't try it. I do not disagree that might be true, but that does not make the experiment irrelevant or unimportant. Of note, the lack of a clear result with micro homeostasis could be linked to the issue above (assumption that quiescence <30 seconds is not sleep - maybe those animals show more short sleep bouts).

3. The authors write "Our observation of spontaneous *C. elegans* quiescence in microfluidic chambers led us to determine whether this behavioral state transition meets the evolutionarily-conserved criteria for classification as sleep: reversibility, a decreased response to stimuli, homeostasis, and a stereotypical posture". They then presented little to no data on posture. This should be added and emphasized more.

4. The data on change in arousal is misleading (Fig 2C). Perhaps the less active/inactive animal just does not move as much because it is less/not active before the stimulus. Can the authors measure how many animals respond to a stimulus by comparing briefly inactive animals (for example, inactive for <5 seconds before stimulus delivery) to those inactive for a time that meet their definition of sleep? This would suffer from the issues I raise above, of course, but by focusing on very brief periods of inactivity prior to the stimulus (even just 1 second), the authors should be able to assess activity changes that start from the same baseline in sleeping vs non sleeping animals.

5. The authors say μ Sleep is similar to SIS, but they find it requires RIS and ALA, and in fact the strongest effect is with RIS. This suggests an overlapping but distinct state. The authors should do a few more experiments to flesh this out. In particular, they show sensory inputs affect sleep, but are these all stress dependent as in SIS? Namely, Is the sleep response changed in *ceh* mutants after temperature or mechanosensory stress? In the current manuscript the authors only test the sensory mutants in those assays.

6. In Fig 6 – can the authors show how sensory inputs change baseline activity? Is activity always the inverse of sleep or are these dissociable? There is much richer data that was acquired and could be used to provide more detail for these experiments.

7. A clear opportunity for a new biological finding would be to show how the different sensory stressors converge to induce sleep. Can the authors define a neuron upstream of ALA/RIS that transforms sensory stimuli regardless of type into a sleep signal, using whole brain imaging approaches? Or, related to point 6, can they show that either RIS or ALA become active (using approaches in Fig 5)?

Reviewer #3 (Remarks to the Author):

In their paper " μ Sleep: A Spontaneous *C. elegans* Sleep State Regulated by Satiety, Thermosensation and Mechanosensation" Robinson et al develop microfluidic assays and characterize the effects of different environmental conditions (food, temperature, mechanical stimuli) on sleep amount. The type of sleep the authors have studied is convincingly shown to present a sleep state, as it fulfills at least some of the behavioral criteria that define sleep (reversibility and reduced arousal threshold, (Raizen et al. 2008)) and as it is controlled by the two key neurons that are known to control other types of sleep in *C. elegans* (ALA (Buskirk et al. 2007) and RIS (Turek et al. 2013)).

Overall, the present paper presents a set of observations and tools that are interesting to the *C. elegans* sleep field. The microfluidic system presents an addition to the existing zoo of microfluidic

sleep study devices. However, the authors appear to have studied a type of sleep that is related to the type of sleep that has been studied before, triggered by an overlapping set of cues such as stress (Hill et al. 2014), satiety (You et al. 2008), and sensory stimulation. The type of sleep described here appears to be a mix of the previously RIS-induced sleep that is present during lethargus and other stages (Turek et al 2013, Wu et al. 2018) and the stress-induced sleep mediated by ALA (Buskirk et al. 2008, Hill et al. 2014). These types of sleep have been shown previously to be highly plastic and environmental conditions can change sleep amount dramatically (You et al. 2008, Skora et al. 2018, Wu et al. 2019, de Bardeleben et al. 2018, Churgin et al. 2017, ...). The argument that μ Sleep is a novel type of sleep is poorly supported and the fact that this type of sleep is induced by the canonical sleep neurons shows that μ Sleep is a canonical type of sleep. Thus, based on the data presented, it does not seem justified to brand a new name for sleep here, i.e. introducing a new term, μ Sleep, is rather misleading to the field.

The value of this manuscript lies in the characterization of how sensory inputs modulate sleep amount. The control of microfluidics sleep by sensory and interneuron circuits is interesting. The demonstration that AFD is a major sensory neuron mediating the effects of temperature on sleep is a valuable addition to the literature. The role of starvation/food availability on sleep has been shown by several previous publications (You et al. 2008, Skora et al. 2018, Wu et al 2019) but the role of mechanosensation in controlling sleep has not been studied much except two studies, one suggesting that ALA could act as a mechanoreceptor but not showing that mechanostimulation could cause sleep (Sanders et al. 2013) and one that showed that mechanical stimulation activates the RIS neuron (Spies et al. 2018).

Overall, the paper will present a valuable addition to the sleep literature in the *C. elegans* field and will be suitable for publication if some issues are addressed. The manuscript can be modified mostly by editing and extracting additional data from the existing experiments and there are no major additional experiments necessary, only minor experiments. There are the following points that should be addressed:

Major points:

1

The authors claim to have found a new type of sleep in *C. elegans* microfluidic devices they called μ Sleep. In *C. elegans*, sleep has been found during various conditions and stages. The new name suggests that μ Sleep presents a unique and novel type of sleep. This conclusion is based on quantitative parameters. However, previous work from several labs has shown that sleep parameters are highly plastic and depend on external conditions (You et al. 2008, Skora et al. 2018, Wu et al. 2019, de Bardeleben et al. 2018, Churgin et al. 2017, ...). Thus, the type of sleep studied here falls within the range of observed sleep patterns. Also, the authors show that μ Sleep is controlled by the same neurons that were previously described to control sleep in *C. elegans* (ALA (Buskirk et al. 2007) and RIS (Turek et al. 2013)). Hence, the sleep behavior the authors observed most likely presents the sleep state that is present in *C. elegans* and that is under a plethora of control pathways. Introducing a novel term, i.e. μ Sleep, is not justified and misleading, particularly for non-specialists of the *C. elegans* sleep field. The authors should find a better term that reflects that the sleep that they are studying is on a continuum of the previously described sleep spectrum. In Figure 1 the authors call their behavior microfluidics-induced quiescence. Accordingly, a term such as microfluidics-induced sleep would seem appropriate. Alternatively, terms such as temperature-controlled sleep or mechanical stress-induced sleep would also be appropriate descriptors.

2

The pan-neuronal imaging data is rather shallow compared to previous pan-neuronal studies of sleep (Skora et al 2018 and Nichols et al. 2017). Figure 4 shows traces for ~ 10 neurons ($\sim 3\%$ of all neurons) and their identities are not shown. Previous work from the Zimmer lab (Skora et al 2018 and Nichols et al. 2017) has identified a much higher number of neurons in their recordings during sleep showing that it is possible to extract not all but a larger fraction of neurons from paralyzed worms. Being able to image from non-paralyzed, only physically restrained, worms would present an advantage over the previous methods but clearly is more challenging. Nevertheless, it would be helpful if the authors could extract more neurons from their recordings they have already performed and provide the neuron identifiers for these neurons. Imaging a

single confocal plane probably cannot represent whole brain imaging. The authors make a strong statement in the abstract that a major advantage of their system is to do whole-brain imaging. Thus, taking stacks through the whole brain and extracting a substantial number of neurons would present an appropriate proof-of-concept experiment for whole brain imaging, even if the data sets are not analyzed in detail. This experiment should be doable with the existing system by taking stacks.

Minor

3

A major selling point of this manuscript is that this type of sleep can be studied in microfluidic devices. The devices are sophisticated and allow certain manipulations, but most studies of sleep in *C. elegans* have been carried out in microfluidic structures, as such experiments are difficult to do without such devices (wormotel (Churgin et al. 2017, simple PDMS chambers (Singh et al. 2011, Huang et al 2017), agarose microchambers (Bringmann 2011), drops (Belfer et al 2013), ...). The authors use their microfluidic device for pan-neuronal imaging, which is an interesting approach to study sleep. However, worms need to be almost completely immobilized inside a microfluidic channel, which the authors show affects sleep behavior. Also, the biological relevance of microfluidic device-induced sleep is not clear but my guess would be that this behavior presents types of sleep that are present also outside of such devices but have not been measured due to technical limits. It would be helpful to discuss these experimental limits and place them a bit more into the context of the existing assays using microfluidic devices.

4

In Fig 2 reversibility is tested for by measuring an increase in activity upon sensory stimulation. Since activity generally is increased upon blue light stimulation it would be best to extract also immobility from these data sets as a measure for sleep. In C: could the baseline condition (before the stimulus) be shown as well? These data could be extracted from the existing data sets.

5

Two pieces of literature that might help interpret the effects of mechanosensation on sleep amount: It has been suggested that ALA acts as a mechanoreceptor, could thus ALA mediate the effects of mechanical stimulation of sleep amount (Sanders et al 2013, PMID:24341457)? Also, mechanical stimulation has been shown to increase RIS activity (Spies et al. 2018, PMID: 29950594).

6

The paper emphasizes the behavioral criteria that define sleep, which include homeostasis. Traditionally, homeostasis is tested by sleep deprivation, which should result in an increased sleep amount or depth. This was first shown for *C. elegans* using sensory stimulation (Raizen et al. 2009). If the authors could provide such an experiment that would be helpful. For example, the worm could be kept awake for a defined period of time using blue light or mechanical stimulation as possible with the microfluidic devices developed by the authors and the subsequent sleep response could be quantified. A relatively simple experiment would seem sufficient to make the point.

We thank the reviewers for their fair and thorough comments on our original work. Based on these comments, we have completed new analyses of existing data and performed new behavioral and neural imaging experiments. We believe that this additional work both addresses the reviewers concerns and makes for a stronger manuscript.

Below is a point-by-point response to each of the reviewer's comments.

EDITORIAL COMMENTS

1. “We will require that you **add new experimental data** to address the points raised by...”

A. “Reviewer 1 regarding additional characterisation of the neural correlates”

In our response to Reviewer 1 we performed **additional experiments** to track calcium dynamics from over 50 neurons in the animal's head during transitions between wakefulness and putative sleep states. We have added a discussion of these data to text as well as a new figure (see below):

*“We envision microfluidic-induced sleep as a platform to understand the neural circuits and correlates driving brain-state transitions. To highlight this potential, we performed additional experiments to record from more individual neurons during whole-brain imaging. As shown in previous works, tracking individual neurons in a moving *C. elegans* brain is a laborious and challenging process^{66,68}. Following examples from other groups^{32,33,65,69,70}, we circumvented this challenge by chemically paralyzing animals, performing z-scanning, and capturing the activity of 40-100 neurons simultaneously during volumetric imaging (**Supplementary Figure 8**). In a fraction of animals, we observed brain-states that resembled what would be expected from a sleeping animal (decrease in calcium activity in all but a few neurons, see **Figure 4**). However, in this paralyzed preparation we cannot unequivocally claim that these states correspond to sleep because we cannot correlate the calcium imaging with animal behavior. Nevertheless, our data is consistent with our expectations for how paralysis should affect the sleep state given the role of mechanoreception. We found that the probability of observing a putative sleep state during a 10 minute experiment dropped from a rate of ~25% in behaving animals to <5% in paralyzed animals. This result is consistent with our conclusion that mechanosensation of the environment facilitated by worm movement is a primary driver of microfluidic-induced sleep. Though we are limited here by a relatively slow imaging rate (< 3 vol/s) and poor z-sectioning with epifluorescence imaging, these experiments show a path towards using microfluidic-induced sleep to understand how neural circuits drive spontaneous brain-state transitions. Future directions should explore high-resolution rapid volumetric*

imaging techniques⁸¹ to measure both the calcium activity and neuron location (e.g. using GCaMP6s and RFP⁶⁶⁻⁶⁸), to not only capture brain-wide activity but also enable single-neuron mapping to the *C. elegans* connectome.” (pp. 24-25).

Supplementary Figure 8. Whole-brain imaging in paralyzed animals. Representative volumetric imaging data from four paralyzed animals during whole-brain imaging. The majority of the data resembles previous imaging work during wakefulness^{32,33}, wherein many neurons show correlated calcium dynamics. Putative sleep states (top two panels) are labeled but cannot be confirmed

without behavioral readouts. These data demonstrate that microfluidic-induced sleep can be used as a model behavior for understanding how brain-wide neural circuits drive spontaneous brain state transitions.

B. “As well as points 1 and 2 by reviewer 2 regarding assessment of shorter bouts of inactivity, and assessment of sleep deprivation.”

In our original report, we only analyzed sleep bouts lasting longer than 30 s to reduce false detections. Reviewer 2 suggested that this choice may affect the reported trends and conclusions of the manuscript. To address this point, we performed **new analyses** and showed that our conclusions were not dependent on this choice of threshold:

“Furthermore, while these behavioral data only represent sleep bouts lasting longer than 30 s to reduce false detections (see Methods), these results—and others throughout this report—were not dependent on this choice of analyses (**Supplementary Figure 3**).” (pp. 12).

Supplementary Figure 3. Reported sleep trends are unaltered by the choice of minimum sleep time. For all data reported, low-activity bouts lasting less than 30 s were removed in order to reduce false-positive detections (see Methods). Here, we show that reducing this value to even 1 s does not change the reported conclusions. Data sets analyzed here are from various portions of the

manuscript. (A) Genetic mutant data reported in Figure 2 (chamber widths are 50 μm). (B) Satiety data from Figure 6 (chamber widths are 500 μm). (C) Temperature data for WT animals in Figure 6 (chamber widths are 500 μm). (D) Data from WT animals in which chamber width was changed in Figure 6 (chamber widths vary and are indicated in the legend). As expected, data from large chambers (500 μm width) is highly stable due an easy distinguishing between sleep and wakefulness. More variation is seen in smaller chambers (50 μm width), where animal movement is heavily restricted, and animals exhibit more low-activity bouts that are not necessarily sleep behavior. In all cases however, the reported trends in throughout the manuscript are sound.

In our original report, we showed that animals in microfluidic-induced sleep exhibited both reversibility and a decreased response to stimuli. Reviewers 2 and 3 made the suggestion that we should also directly test for homeostasis following sleep deprivation, even though sleep deprivation can lead to a stress response--thus making homeostatic mechanisms ambiguous. To assess sleep deprivation, we performed **new optogenetic experiments** to inhibit the RIS neuron and control for effects due to stress:

*“To directly test the effects of sleep deprivation, we performed optogenetic inhibition of the RIS neuron (**Supplementary Figure 2B-D**), which is known to be implicated in *C. elegans* sleep^{38,39,43} (see also **Figure 3** and **Figure 5**). This optogenetic approach allowed us to have a control group (raised in the absence of all-trans retinal) that is also exposed to light but does not have optogenetic inhibition of the RIS neuron, thus controlling for any potential effects of stress produced by blue light illumination. We found that although RIS inhibition for 30 min did not fully abolish sleep, optically inhibited animals exhibited 324% more sleep and 230% more low-activity behavior compared to control animals during the refractory period following the optogenetic inhibition of the RIS neuron ($p < 0.01$) (**Supplementary Figure 2D**). RIS-inhibited animals also showed sleep bouts lasting 1.5 times longer than control animals ($p < 0.01$) (**Supplementary Figure 2C-D**). From our results combined, we conclude that animals show a homeostatic rebound in response to prolonged RIS inhibition. Therefore, microfluidic-induced quiescence indeed meets the behavioral precedents to be called a *C. elegans* sleep state: stereotyped posture, reversibility, a decreased response to stimuli, and homeostasis.” (pp. 11-12).*

Supplementary Figure 2. Evidence for homeostasis in microfluidic-induced sleep. (A) Sleep bout length is dependent on the length of the previous wake bout. (Left) Data points represent individual sleep and wake bout pairs from data collected from 50 and 500 μm chambers with WT animals (see Figure 1, 4 hr-long recordings and Figure 6, 2 hr-long recordings). (Right) Wake bout lengths binned into 5 min windows. As the wake bout length increases, the following sleep bouts increase in length, suggesting a micro-homeostatic mechanism (ns = not significant, **** $p < 0.0001$ compared to 0-5 min data set, Kruskal-Wallis with a *post-hoc* Dunn-Sidak test). (B-D) Animals showed a homeostatic rebound in response to prolonged RIS optogenetic inhibition. (B) (Left) Heatmaps of animal behavioral activity during an optogenetic assay where green light illumination was delivered between 1-1.5 hr. (Right) Raster plots of detected sleep bouts. +ATR animals were

grown with OP50 seeded with *all-trans retinal*. -ATR animals were not, thus making optogenetic inhibition ineffective. (C) We used a threshold 66% higher than the sleep threshold to detect all low-activity bouts (these include sleep and other low-activity behaviors). +ATR animals showed similar behavioral activity during the stimulus, but more low-activity after the stimulus, suggesting that prolonged RIS inhibition led to a homeostatic response following optogenetic inhibition of the sleep-promoting RIS neuron. (D) Using our standard threshold for sleep detection, +ATR and -ATR animals surprisingly showed no significant difference in total sleep during illumination. However, +ATR animals showed significantly more sleep following stimulation, suggesting a homeostatic response to prolonged RIS inhibition. In addition, +ATR animals showed longer sleep bouts following illumination. These results suggest that *C. elegans* show increased microfluidic-induced sleep and low-activity following extended RIS inhibition. (ns = not significant, * $p < 0.05$, ** $p < 0.01$, unpaired two-sided t-test).

2. “We will also require that you revise the term “*mu-sleep*” used to describe this behavioural phenomenon, in line with suggestion from reviewer 3, to avoid suggestion that this is a novel sleep state.”

Throughout the manuscript we now refer to the observed sleep behavior as merely “microfluidic-induced sleep,” as suggested by Reviewer 3.

REVIEWER 1

Remarks to authors:

*“In this manuscript Gonzales and colleagues report so far undescribed experimental conditions that promote sleep in adult *C. elegans*. The best described sleep state in *C. elegans* is developmentally timed sleep (DTS), but during recent years multiple labs have established additional sleep paradigms, e.g. stress and starvation induced sleep. The authors show here that constraint in microfluidic devices promotes spontaneous bouts of immobility, termed here mu-sleep. They validate that mu-sleep fulfills behavioral criteria of sleep (reversibility, increase in arousal threshold, homeostasis). Imaging neuronal activity via a Ca^{++} -indicator, either pan-neuronally expressed or specifically in the sleep promoting neuron RIS they show that mu-sleep likely exhibits the same neuronal activity signature that was previously reported for DTS. Using genetics, the authors show that mu-sleep requires genetic pathways that were previously shown to be required for DTS and stress induced sleep.*

Finally, some evidence is provided for the implication of mechano-receptors in the induction of mu-sleep. Most experiments are sophisticated and statistical analyses are sound. One exception are the brain wide imaging experiments, previous work recorded brain wide activity during sleep at single cell resolution, while in the current study wide field epi-fluorescence imaging can only provide a crude estimate about the neuronal activity changes upon mu-sleep onset. However, for the point that the authors want to make here, this seems sufficient to me.

*The paper provides an incremental step towards understanding regulation of sleep in *C. elegans*. Whether mu-sleep is a distinct new sleep state or a mild form of stress induced sleep remains unclear. Moreover, the factors that trigger mu-sleep remain enigmatic. The various constraining conditions used here vary greatly in how they could affect the animals; also, the ethological relevance of mu-sleep is unclear and only vaguely discussed by the authors. The paper is of special interest to the community of *C. elegans* researchers who uses micro-fluidic devices for neuronal imaging under even constraint conditions; the prevalence of a sleep state under these conditions is important for the interpretations of their results. This aspect should be definitely highlighted in the discussion.”*

We thank the reviewer for their thorough comments and insights, which we address point-by-point in detail below. In addition, based on these concluding remarks regarding the relevance to studying *C. elegans* behavior in microfluidics, we added the following text to the discussion:

*“Importantly, our results show the drastic effects of microfluidic environments on *C. elegans* behavior. Microfluidic assays are commonly used to study nematode biology including, aging, behavior, and neurobiology^{82–84}. Sleep in *C. elegans* is also commonly studied by confining animals to microfluidic chambers^{52,53,56,85–87}. Our data show that the conditions of these micro-environments can strongly affect the sleep phenotype and thus the microfluidic chambers should*

be carefully designed and their effects should be considered when interpreting the results of these experiments.” (pp. 26).

Major comments:

1. *“I think it is likely that flushing worms into the microfluidic devices could cause some mechanical damage. Is mu-sleep simply stress induced sleep in response to this possible damage?”*

We agree that it is very important to ensure that sleep in microfluidic chambers is not simply due to stress that the animals may experience as they are loaded into microfluidic devices. To address this possibility, we performed **new experiments** and made revisions to the main text that show that loading the animals into microfluidic chambers, and the drastic change of environment, is not sufficient to drive the stress-induced sleep response:

*“Finally, we also ruled out the possibility that the process of loading animals into the microfluidic device or changing the fluidic environment leads to a stress response that drives stress-induced sleep. We fabricated large microfluidic chambers that are expected to mimic an open volume of liquid (1200 μm long, 500 μm wide, 90 μm tall), and observed that animals rarely exhibited sleep behavior despite undergoing the same loading process (**Supplementary Figure 7G-I**). Therefore, the stress associated with the loading process and change in the fluidic environment alone is not sufficient to drive stress-induced sleep. In fact, reducing this chamber height to 55 μm reestablished the sleep behavior, further strengthening our claims that the mechanical environment is a strong driver of microfluidic-induced sleep (**Supplementary Figure 7G-I**).” (pp. 23-24).*

Supplementary Figure 7. Waste buildup, chamber volume, and stress from loading do not strongly influence microfluidic-induced sleep. (A) Image of animals confined in microfluidic chambers designed for constantly flowing buffer to stabilize O₂ concentrations and remove the buildup of CO₂ and other byproducts. Blue paths indicate the direction of flow for a single chamber. Flow rate was ~1 mL/hr. (B) Raster plots of detected sleep with and without flow using the geometry in (A). (C) We surprisingly observed more sleep with the buffer flow, indicating microfluidic-induced sleep is likely not driven by changing gas concentration levels biological byproducts (n = 24 for each condition). (D) Chamber designs for maintaining a constant animal compression while changing fluidic volume. Fluid is false-colored in pale blue. (E) Raster plots of detected sleep for each chamber type. (F) Animals in chambers of different volume do not show different amounts of sleep (n = 32 for each condition). (G) Schematic of chamber cross section that have same width and length, but different chamber heights. (H) Raster plots of detected sleep. (I) Sleep in the 90um tall chambers is essentially abolished, demonstrating that the animal loading process and microfluidic environment alone do not drive sleep (n = 22). Sleep dramatically increases in the 55 um tall chambers (n = 50), again demonstrating that the mechanical environment regulates sleep strongly. (ns = not significant, **p < 0.01, unpaired two-sided t-test).

Minor comments:

2. *Results section: “is typically defined (...)”: this sentence seems to suggest that there is a specific reason to have different criteria based on whether it is adult sleep, which I don’t think it’s justified.*

We thank the reviewer for the suggestions and reworded this sentence:

“However, reports of sleep in adult worms have only observed reversibility and a decreased response to stimuli^{40,46}. Here, we tested for a stereotyped posture, reversibility, decreased response to stimuli, and a homeostatic rebound.” (pp. 8).

3. *“Fig. 1: I have big doubts about the very first sentence of the legend/title: Basically, would like to know whether the onset of sleep maybe simply correlates with the volume of the different chamber/well (by testing a couple more chambers with intermediate and/or larger volumes), and/ or the number of times the animal “touches” every wall, before falling into immobility. Very interestingly, it seems that one possible explanation for small chamber, large chamber and worMotel-phenotypes is simply a scaling of the very same reaction: the worms try for some time to move (sampling the liquid environment around it and/or the walls), until it realizes it is in an unescapable environment, with no interesting stimuli around. In such environments, trying to move and expending energy is pointless. The realization takes longer if the environment is larger. Could the duration of the Q bouts be affected by how many times the worm bump or is drafted into the walls (leading to a stimulation that interrupts sleep, waking up the worm)? If this is the case, this would be the simple reason why the Q bouts are longer in in the worMotel. It seems like only very large body movements would be caught while imaging in the smallest chamber, of 50um (there is a video where the worm slightly moving its tail is labelled as sleeping). Might this be the case? So: How would the Q bouts, labelled based on the high RIS activity, compare with the behavioral estimate based on frame subtraction?”*

We thank the reviewer for their comments. We made both edits and performed **new experiments** to address these points.

Rather than the term “phenotypically distinct,” we changed the title of Figure 1, which shows quantitative differences between *C. elegans* quiescence under different conditions:

“Quiescence dynamics are strongly affected by microfluidic environments.” (pp. 7).

We also revised existing text to highlight that microfluidic-induced sleep, stress-induced sleep, and swimming-induced quiescence may be the same behaviors:

*“Thus, we consider microfluidic-induced quiescence to merely display quantifiably distinct dynamics compared to other reported quiescent behaviors in *C. elegans* adults and is not necessarily a new behavioral state.” (pp. 6).*

We also performed **new experiments** with different microfluidic chamber geometries to show that the fluidic volume of the chambers does not dramatically affect sleep (see **Supplementary Figure 7D-F** above):

*“In another experiment, we considered the possibility that the small volume of liquid in different microfluidic chambers could play a role in driving more sleep than through the buildup of gasses or other products. To test this hypothesis, we developed a geometry that maintains the same animal restraint but varies the fluidic volume in the worm chambers (**Supplementary Figure 7D**). The total sleep in these chambers was not significantly different (**Supplementary Figure 7E-F**). These experiments suggest that indeed the mechanical environment is a much stronger driver of sleep than fluidic volume, and not the buildup of metabolic products.” (pp. 22).*

To include the potential role of animal collisions with the chamber walls, we added to the discussion:

*“...the behavioral dynamics of quiescence that we measured is quantitatively distinct from these previously reported worm quiescent behaviors (**Figure 1**), although it is possible that changes in sleep onset time and bout duration in microfluidics is affected by collisions with the chamber walls or other physical cues.” (pg. 25).*

3. (cont'd) “[Fig. 1] Legend: HS is defined as “first hour of heat shock” but is actually half a hour.”

We made a revision to clarify that for heat shock experiments we analyzed the 30 min in which heat was applied *and* the 30 min after heat was removed:

*“For our final comparison point, we heat-shocked animals in the large microfluidic device for 30 min at 30 °C to induce stress-quiescence, then analyzed animal behavior during the heat shock plus a 30 min recovery period⁴⁰ (**Figure 1B**).” (pg. 5).*

4. “Fig. 2: Why so different n numbers for the 2C?”

Animals received mechanical stimulation every 3 min during the experiment. During post-processing, we then classified animals as being either in the wake or sleep state. Animals are simply more likely to be awake, which leads to more data for this condition. We highlight this point in the Methods:

“Note that because animals are more likely to be in the wake state, this classification method leads to lower n-values for the sleep condition.” (pg. 33).

5. *“Fig. 4: Not clear whether the neurons quantified for their activity in C are the same random 10 neurons picked for tracking and for detecting movement (see Methods).”*

We revised the figure legend to clarify this point:

“The neurons chosen for analysis were randomly selected from the field-of-view and are not necessarily the same neurons across every animal.” (pp. 15).

6. *“Fig.5A: I suggest to make an inset zooming in on the heads. Fig.5B: Shaded regions are invisible upon print.”*

We thank the reviewer for this suggestion and have included a zoomed inset of the RIS neuron and darkened the shaded regions in Fig. 5:

7. *“Fig. 6: Legend: “no food IN the buffer” (“in” is missing)”*

Thank you for the edit, we made the following change to the Figure 6 caption:

“‘Baseline’ indicates the standard experimental conditions: 500 μm chamber width, no food in the buffer, and a 22 $^{\circ}\text{C}$ temperature.” (pg. 19).

8. “Fig. Suppl. 3: Panel labelling goes A-C-D”

Thank you for the edit, we made this change to the Supplementary Figure:

9. “Methods: Why for RIS imaging the threshold for sleep bout detection is 60s? It was 30s for both behavior and whole-brain imaging.”

Thank you for catching this typo, we made the correction:

“After normalizing behavioral activity, a threshold of 0.2 was used to detect sleep bouts, with a minimum sleep time of 30 s.” (pp. 35).

10. “Methods: DAF-16(...) referring to Suppl. Fig1 wrongly (it’s Suppl. Fig.3)”

We thank the reviewer for their comments and made this correction (correcting for new additional Supplementary Figures):

“The puncta count reported (Supplementary Figure 6)...” (pg. 37).

REVIEWER 2

Remarks to authors:

*“In this manuscript by Gonzales and colleagues, the authors describe a new platform for studying sleep in *C. elegans*. They demonstrate that μ Sleep is phenotypically and mechanistically similar to SIS (and DTS), and the microfluidic approach provides opportunities for studying sleep mechanisms both with regard to behavioral detail and imaging. They go on to show that μ Sleep is modulated by diverse sensory inputs, including temperature and mechanical stress. The work is interesting and describes an important new platform, though does not reveal much in the way of new sleep biology.”*

Major comments:

1. *“I am not convinced with how the authors define what they count as sleep (quiescence >30 seconds). This is a critical issue and should be discussed prominently in the results, with evidence for how this was selected....if included shorter bouts simply increases noise without altering the overall patterns observed in a few of the experimental paradigms, I would be more convinced.”*

We thank the reviewer for the comment and addressed this point with new analyses (see response to Editor’s comments and **Supplementary Figure 3** embedded above). We re-analyzed multiple existing data sets across the manuscript with varying experimental conditions, microfluidic chambers geometries, and animal strains. We found that **reducing the minimum sleep threshold to even 1 s did not change the reported trends.**

2. *“The authors need to test sleep deprivation directly. If the animals do not exhibit homeostatic rebound, I do not believe it would invalidate the model. But in the current version, they say only that sleep deprivation can produce a stress response so they don't try it.”*

To address this point, we performed **new experiments** using optogenetic inhibition of the RIS neuron (see response to Editorial comments and **Supplementary Figure 2** embedded above). Although optogenetic inhibition of the RIS neuron appeared to have a relatively weak effect on the observed behavior during stimulation, we observed a significant increase in both sleep and low-activity behavior compared to a control group during the refractory period following stimulation. **These results suggest that animals show a homeostatic rebound in microfluidic-induced sleep following an extended period of RIS inhibition.**

3. “The authors write ‘Our observation of spontaneous *C. elegans* quiescence in microfluidic chambers led us to determine whether this behavioral state transition meets the evolutionarily-conserved criteria for classification as sleep: reversibility, a decreased response to stimuli, homeostasis, and a stereotypical posture.’ They then presented little to no data on posture. This should be added and emphasized more.”

To address this point, we **analyzed** existing data and found a significant increase in animal body curvature during sleep compared to wakefulness:

“Similar to previous reports^{57,59,60}, we found that animals exhibited a stereotypical posture during quiescence (**Figure 2A**). The body curvature of animals crawling on agar typically decreases during developmental sleep⁵⁷. In our case, we found that animals swimming in microfluidics show increased body curvature during sleep (from 3.9 ± 0.04 radians during wakefulness to 4.5 ± 0.1 radians during quiescence (mean \pm sem), $p < 0.0001$, **Figure 2A, right**).” (pp. 8).

Figure 2. During microfluidic-induced sleep animals display stereotypical posture, reversibility, and a decreased response to weak stimuli. (A) *C. elegans* show increased body curvature during sleep. (Left) Representative example of quantified behavioral activity and normalized body curvature (see Methods) for a single animal during a 2 hr recording. Dotted line indicates sleep threshold. Increased body curvature correlates with sleep bouts. (Right) Across a population of animals, average body curvature increases during sleep (from “Large μ Fluidic” data set in **Figure 1**, $n = 47$ animals, **** $p < 0.0001$, unpaired t-test). (B) The sleep state is reversible. (Left) Heatmaps

of animal behavioral activity (top) and nose speed (bottom). Measurements begin with all animals in the quiescent state. The 5 s light pulse at $t = 30$ s results in a rapid increase in animal activity and nose speed. (Right) Mean behavioral activity (top) and nose speed (bottom) of each animal during the first and second 30 s of imaging shows a clear transition from sleep to wake states. Dotted line indicated sleep threshold. ($n = 13$ worms, **** $p < 0.00001$, paired t-test). (C-E) Quiescent animals have a decreased response to weak sensory stimuli. (C) Heatmaps of activity from awake and sleeping animals that received strong or weak mechanical stimuli from microfluidic valves (**Supplementary Movie 4-Supplementary Movie 5**). For each condition, only the 25 trials with the highest mean activity post-stimulation are shown. “Wake” indicates trials in which animals were awake but have a below-average activity before stimulation (see Methods). “Sleep” indicates trials in which animals were quiescent before stimulation. Heatmaps contain an ~ 2 s long stimulation artifact beginning at 10 s due to movement from microfluidic valves. (D) Mean behavioral activity before and after mechanical stimulation from the trials in (C). Dotted line indicates sleep threshold. In all cases, average activity significantly increases after the stimulation (largest p -value = 0.002, paired two-sided t-test). However, following the stimulation behavioral activity is not significantly different in all cases other than when quiescent animals received weak mechanical stimuli (Error bars are sem; Strong-Wake $n = 108$, Strong-Sleep $n = 51$, Weak-Wake $n = 176$, Weak-Sleep $n = 29$; **** $p < 0.0001$, ns = not significant, Kruskal-Wallis with a *post-hoc* Dunn-Sidak test). (E) Fraction of animals in the wake state following mechanical stimuli. A significant difference only occurs in the case of quiescent animals receiving a weak mechanical stimuli. (Error bars are standard deviation, calculated by bootstrapping each data set with 5000 iterations; ns = not significant, **** $p < 0.00001$; significance was calculated by data resampling 5000 iterations and a *post hoc* Bonferroni correction).

4. “The data on change in arousal is misleading (Fig 2C). Perhaps the less active/inactive animal just does not move as much because it is less/not active before the stimulus. Can the authors measure how many animals respond to a stimulus by comparing briefly inactive animals (for example, inactive for < 5 seconds before stimulus delivery) to those inactive for a time that meet their definition of sleep? This would suffer from the issues I raise above, of course, but by focusing on very brief periods of inactivity prior to the stimulus (even just 1 second), the authors should be able to assess activity changes that start from the same baseline in sleeping vs non sleeping animals.”

We revised the text and re-analyzed data to further test for decreased arousal. As suggested by the reviewer, our new analysis showed that some animals originally classified as awake display very short low-activity bouts immediately prior to the stimulus. We then compared these “Low-Activity” animals to animals in the wake and sleep states (**Supplementary Figure 1**).

Supplementary Figure 1. Further comparison of behavioral responses to weak mechanical stimulation. (A) (Left) Heatmap of all behavioral activity from all wake animals receiving a weak mechanical stimulus ($n = 176$ total trials, note that in Figure 2 only the first 25 animals are shown for clarity). (Right) We further parsed this data set into “Wake” and “Low-Activity” states, where “Low-Activity” is defined as having an average behavioral activity level below the sleep threshold for a short 3 s period immediately before the stimulus. (B) Average behavioral activity before and after the weak stimulus for animals in the Wake and Low-Activity states prior to the stimulus. We also compared these results to animals in the Sleep state prior to the stimulus (same data as Figure 2D, error bars are sem, $***p < 0.001$, Kruskal-Wallis with a *post-hoc* Dunn-Sidak test). (C) Fraction of animals awake following the stimulus for the Wake and Low-Activity states. We also compared these data to the Sleep state (same data as Figure 2E). Animals in the Sleep state are less likely to respond to weak stimuli and transition to wakefulness, compared to animals in the wake state regardless of their activity level prior to the stimulus. (Error bars are standard deviation, calculated by bootstrapping each data set with 5000 iterations; ns = not significant, $***p < 0.001$, $****p < 0.0001$; significance was calculated by data resampling 5000 iterations and a *post hoc* Bonferroni correction).

In addition, to give a measure of the number of animals responding to stimulation, we further quantified existing data and calculated the fraction of animals in the wake state following mechanical stimuli (see Figure 2E, Supplementary Figure 1C embedded above). Finally, we

also revised the text to emphasize that mechanical stimuli evoked a strong behavioral response in all cases other than a weak stimuli to quiescent animals. For example, a strong stimulus to quiescent animals showed that even low-activity animals can robustly respond to external stimuli. We found that animals in the Sleep state are less likely to respond to weak stimuli and transition to wakefulness, compared to animals in the Wake state regardless of their activity level prior to the stimulus. These new analyses and revisions can be found in **Figure 2E**, **Supplementary Figure 1C**, and revised text:

“When we applied a weak stimulus to wake animals (low pressure, 15 psi), we again reliably recorded a significant increase in activity that matched the response of the strong stimulation (Figure 2C, Supplementary Movie 5). However, when we delivered a weak stimulus to quiescent animals, they responded weakly, exhibiting an average behavioral activity less than half that of animals in the awake state (Figure 2C-D, Supplementary Movie 5). In fact, less than 40% of quiescent animals that received a weak stimulus transitioned to wakefulness, compared to >75% of animals being in the awake state following stimulation for all other experimental conditions (Figure 2E). Importantly, because the strong stimuli evoked a similar strong behavioral response from both quiescent animals and awake animals, animals in the quiescent state are not less capable of responding to stimuli provided the stimuli is sufficiently intense (Figure 2C-D). Further analyses also showed that animals in the sleep state are less likely to respond to weak stimuli and transition to wakefulness, compared to animals in the wake state regardless of their activity level prior to the stimulus (Supplementary Figure 1). When we analyzed data from the wake animals, we found that nearly 70% of animals with activity levels below our sleep threshold for 3 seconds prior to the weak stimuli transitioned to wakefulness, compared to <40% for animals that we classified to be in the sleep state prior to a weak stimulus (Supplementary Figure 1). These results are consistent with quiescence being a sleep state rather than simply a low-activity state of wakefulness^{25,26,40,61}.” (pp. 8-9).

5. *“The authors say μ Sleep is similar to SIS, but they find it requires RIS and ALA, and in fact the strongest effect is with RIS. This suggests an overlapping but distinct state. The authors should do a few more experiments to flesh this out. In particular, they show sensory inputs affect sleep, but are these all stress dependent as in SIS? Namely, Is the sleep response changed in *ceh* mutants after temperature or mechanosensory stress? In the current manuscript the authors only test the sensory mutants in those assays.”*

We agree with the reviewer that many questions still remain concerning microfluidic-induced sleep and environmentally-driven *C. elegans* sleep in general. A significant amount of work must be done to fully untangle the role of sensory circuits, RIS, and ALA on microfluidic-induced sleep. However, we also recognize that the number of possible experimental conditions, mutant strains, and potential calcium-imaging targets are quite vast to fully elucidate these circuits. We

have expanded our discussion section to highlight these future endeavors in understanding *C. elegans* sleep circuits with respect to microfluidic environments:

“Many questions remain with respect to C. elegans sleep and the role of sensory inputs in driving sleep behavior. For example, sleep-promoting neurons such as ALA and RIS drive behavioral quiescence, but what ensemble activity occurs upstream of these circuits? The interaction between multiple sensory inputs, the ALA/RIS neurons, and diverse behavioral outputs are likely highly complex, non-linear systems involving multiple pathways and neuromodulators^{34,38,43–45,53}. Future work will help answer these questions and determine the relationship between ALA and RIS and whether these neurons are critical for driving sleep in all the environmental conditions tested in this report (Figure 6). Furthermore, ALA and RIS themselves are known to respond to mechanical stimulation^{86,89}, potentially providing a mechanism for why animal confinement drives sleep behavior. These “sleep-driving” neurons that also act as sensory neurons are a prime example of the multimodal nature of the C. elegans nervous system, which adds another element of complexity when dissecting the sensory and sleep circuits involved in environmentally-driven sleep.” (pp. 27-28).

6. *“In Fig 6 – can the authors show how sensory inputs change baseline activity? Is activity always the inverse of sleep or are these dissociable? There is much richer data that was acquired and could be used to provide more detail for these experiments.”*

To address this point we analyzed our existing data set and found that behavioral activity during wakefulness remained largely invariant to our environmental manipulations and the mutant strains tested. We revised the text in several places and included an additional supplementary figure to show our results:

“We also observed that only in the “Starved” condition did animals show measurably lower behavioral activity during wakefulness (Supplementary Figure 4A).” (pp. 18).

“In addition, temperature had no effect on wake-state behavioral activity for either WT or mutant animals (Supplementary Figure 4B).” (pp. 19).

“While animal restraint indeed changed the measured behavioral activity, these differences in sleep between WT and mec-10 animals cannot be explained by changes in behavioral activity during wakefulness (Supplementary Figure 4C).” (pp. 20).

Supplementary Figure 4. Behavioral activity during wakefulness in different environmental conditions. Data is the averaged wake-state behavioral activity for each animal in Figure 6. (A) Only WT animals in the starved condition only show less activity compared to -Food and +Food. (B) No significant differences were found in either WT or *gcy-23(nj37);gcy-8(oy44);gcy-18(nj38)* mutants (labeled as -AFD) at any microfluidic device temperature. (C) As expected, restricting animal movement significantly changes behavioral activity during wakefulness. However, no changes were observed between WT and *mec-10(tm1552)* animals. (**** $p < 0.0001$, ns = not significant, Kruskal-Wallis with a *post-hoc* Dunn-Sidak test).

7. “A clear opportunity for a new biological finding would be to show how the different sensory stressors converge to induce sleep. Can the authors define a neuron upstream of ALA/RIS that transforms sensory stimuli regardless of type into a sleep signal, using whole brain imaging approaches? Or, related to point 6, can they show that either RIS or ALA become active (using approaches in Fig 5)?”

We agree with the reviewer that this is a clear goal of the microfluidic-induced sleep platform. Ideally whole-brain imaging under different environmental conditions combined with the worm connectome would reveal the interaction of these neural circuits. However, this is a challenging endeavor beyond the scope of this work. In addition to performing **new experiments** where we capture more neurons during whole-brain imaging (see response to the editor and **Supplementary Figure 8** embedded above), we have added a brief discussion of the limitations of our current imaging approach and future directions to achieve these goals:

“Though we are limited here by a relatively slow imaging rate (< 3 vol/s) and poor z-sectioning with epifluorescence imaging, these experiments show a path towards using microfluidic-induced sleep to understand how neural circuits drive spontaneous brain-state transitions. Future directions should explore high-resolution rapid volumetric imaging techniques⁸¹ to measure both the calcium activity and neuron location (e.g. using GCaMP6s and RFP⁶⁶⁻⁶⁸), to not only capture brain-wide activity but also enable single-neuron mapping to the C. elegans connectome.” (pp. 25).

REVIEWER 3

Remarks to authors:

*“In their paper “ μ Sleep: A Spontaneous *C. elegans* Sleep State Regulated by Satiety, Thermosensation and Mechanosensation” Robinson et al develop microfluidic assays and characterize the effects of different environmental conditions (food, temperature, mechanical stimuli) on sleep amount. The type of sleep the authors have studied is convincingly shown to present a sleep state, as it fulfills at least some of the behavioral criteria that define sleep (reversibility and reduced arousal threshold, (Raizen et al. 2008)) and as it is controlled by the two key neurons that are known to control other types of sleep in *C. elegans* (ALA (Buskirk et al. 2007) and RIS (Turek et al. 2013)).*

*Overall, the present paper presents a set of observations and tools that are interesting to the *C. elegans* sleep field. The microfluidic system presents an addition to the existing zoo of microfluidic sleep study devices. However, the authors appear to have studied a type of sleep that is related to the type of sleep that has been studied before, triggered by an overlapping set of cues such as stress (Hill et al. 2014), satiety (You et al. 2008), and sensory stimulation. The type of sleep described here appears to be a mix of the previously RIS-induced sleep that is present during lethargus and other stages (Turek et al 2013, Wu et al. 2018) and the stress-induced sleep mediated by ALA (Buskirk et al. 2008, Hill et al. 2014). These types of sleep have been shown previously to be highly plastic and environmental conditions can change sleep amount dramatically (You et al. 2008, Skora et al. 2018, Wu et al. 2019, de Bardeleben et al. 2018, Churgin et al. 2017, ...). The argument that μ Sleep is a novel type of sleep is poorly supported and the fact that this type of sleep is induced by the canonical sleep neurons shows that μ Sleep is a canonical type of sleep. Thus, based on the data presented, it does not seem justified to brand a new name for sleep here, i.e. introducing a new term, μ Sleep, is rather misleading to the field. The value of this manuscript lies in the characterization of how sensory inputs modulate sleep amount. The control of microfluidics sleep by sensory and interneuron circuits is interesting. The demonstration that AFD is a major sensory neuron mediating the effects of temperature on sleep is a valuable addition to the literature. The role of starvation/food availability on sleep has been shown by several previous publications (You et al. 2008, Skora et al. 2018, Wu et al 2019) but the role of mechanosensation in controlling sleep has not been studied much except two studies, one suggesting that ALA could act as a mechanoreceptor but not showing that mechanostimulation could cause sleep (Sanders et al. 2013) and one that showed that mechanical stimulation activates the RIS neuron (Spies et al. 2018).*

*Overall, the paper will present a valuable addition to the sleep literature in the *C. elegans* field and will be suitable for publication if some issues are addressed. The manuscript can be*

modified mostly by editing and extracting additional data from the existing experiments and there are no major additional experiments necessary, only minor experiments. There are the following points that should be addressed:”

Major points:

1. *“The authors claim to have found a new type of sleep in C. elegans microfluidic devices they called μ Sleep. In C. elegans, sleep has been found during various conditions and stages. The new name suggests that μ Sleep presents a unique and novel type of sleep. This conclusion is based on quantitative parameters. However, previous work from several labs has shown that sleep parameters are highly plastic and depend on external conditions (You et al. 2008, Skora et al. 2018, Wu et al. 2019, de Bardeleben et al. 2018, Churgin et al. 2017, ...). Thus, the type of sleep studied here falls within the range of observed sleep patterns. Also, the authors show that μ Sleep is controlled by the same neurons that were previously described to control sleep in C. elegans (ALA (Buskirk et al. 2007) and RIS (Turek et al. 2013)). Hence, the sleep behavior the authors observed most likely presents the sleep state that is present in C. elegans and that is under a plethora of control pathways. Introducing a novel term, i.e. μ Sleep, is not justified and misleading, particularly for non-specialists of the C. elegans sleep field. The authors should find a better term that reflects that the sleep that they are studying is on a continuum of the previously described sleep spectrum. In Figure 1 the authors call their behavior microfluidics-induced quiescence. Accordingly, a term such as microfluidics-induced sleep would seem appropriate. Alternatively, terms such as temperature-controlled sleep or mechanical stress-induced sleep would also be appropriate descriptors.”*

To address this point and to avoid the suggestion that we have discovered a new C. elegans behavior, as suggested we refer to these quiescent state as “microfluidic-induced sleep” throughout the manuscript.

2. *“The pan-neuronal imaging data is rather shallow compared to previous pan-neuronal studies of sleep (Skora et al 2018 and Nichols et al. 2017). Figure 4 shows traces for ~10 neurons (~ 3% of all neurons) and their identities are not shown. Previous work from the Zimmer lab (Skora et al 2018 and Nichols et al. 2017) has identified a much higher number of neurons in their recordings during sleep showing that it is possible extract not all but a larger fraction of neurons from paralyzed worms. Being able to image from non-paralyzed, only physically restrained, worms would present an advantage over the previous methods but clearly is more challenging. Nevertheless, it would be helpful if the authors could extract more neurons from their recordings they have already performed and provide the neuron identifiers for these neurons. Imaging a single confocal plane probably cannot represent whole brain imaging. The authors make a strong statement in the abstract that a major advantage of their system is to do whole-brain imaging. Thus, taking stacks through the whole brain and extracting a substantial number of neurons would present an appropriate proof-of-concept experiment for whole brain*

imaging, even if the data sets are not analyzed in detail. This experiment should be doable with the existing system by taking stacks.”

We thank the reviewer for the suggestion. As we stated in our responses to the Editor, we believe that our single-plane whole brain imaging experiments (**Figure 4**), single-neuron RIS imaging data (**Figure 5**), in combination to existing literature (Nichols et al. 2017, Skora et al. 2018, Wu et al. 2018) support our original claim that a majority of the *C. elegans* brain shows decreased activity during microfluidic-induced sleep.

However, to show that microfluidic-induced sleep is a pathway towards capturing *spontaneous* brain-state transitions, we performed **new experiments** using volumetric whole-brain imaging in paralyzed animals to capture the activity of 40-80 neurons in the worm head during putative sleep states (see response to the editor and **Supplementary Figure 8** embedded above).

Please also note new inclusions to the text, which discuss the limitations to our current imaging approach, but highlight a viable path towards using microfluidic-induced to understand the neural circuits underlying *C. elegans* sleep and brain-state transitions:

*“Though we are limited here by a relatively slow imaging rate (< 3 vol/s) and poor z-sectioning with epifluorescence imaging, these experiments show a path towards using microfluidic-induced sleep to understand how neural circuits drive spontaneous brain-state transitions. Future directions should explore high-resolution rapid volumetric imaging techniques⁸¹ to measure both the calcium activity and neuron location (e.g. using GCaMP6s and RFP⁶⁶⁻⁶⁸), to not only capture brain-wide activity but also enable single-neuron mapping to the *C. elegans* connectome.” (pp. 25).*

Minor points:

3. *“A major selling point of this manuscript is that this type of sleep can be studied in microfluidic devices. The devices are sophisticated and allow certain manipulations, but most studies of sleep in *C. elegans* have been carried out in microfluidic structures, as such experiments are difficult to do without such devices (wormotel (Churgin et al. 2017, simple PDMS chambers (Singh et al. 2011, Huang et al 2017), agarose microchambers (Bringmann 2011), drops (Belfer et al 2013), ...). The authors use their microfluidic device for pan-neuronal imaging, which is an interesting approach to study sleep. However, worms need to be almost completely immobilized inside a microfluidic channel, which the authors show affects sleep behavior. Also, the biological relevance of microfluidic device-induced sleep is not clear but my guess would be that this behavior presents types of sleep that are present also outside of such devices but have not been measured due to technical limits. It would be helpful to discuss these experimental limits and place them a bit more into the context of the existing assays using microfluidic devices.”*

We thank the reviewer for the insight. To address this we have added the following discussion:

“Importantly, our results show the drastic effects of microfluidic environments on C. elegans behavior. Microfluidic assays are commonly used to study nematode biology including, aging, behavior, and neurobiology^{82–84}. Sleep in C. elegans is also commonly studied by confining animals to microfluidic chambers^{52,53,56,85–87}. Our data show that the conditions of these micro-environments can strongly affect the sleep phenotype and thus the microfluidic chambers should be carefully designed and their effects should be considered when interpreting the results of these experiments. However, although we observe changes in sleep behavior in the context of microfluidics—and most notably during increased confinement—quiescence behavior may increase in standard agar environments as well due to changes in animal confinement. For example, we hypothesize that experiments with agarose-based confinement chambers^{86,88} would observe the similar changes in behavior to those reported here.” (pp. 26).

4. *“In Fig 2 reversibility is tested for by measuring an increase in activity upon sensory stimulation. Since activity generally is increased upon blue light stimulation it would be best to extract also immobility from these data sets as a measure for sleep. In C: could the baseline condition (before the stimulus) be shown as well? These data could be extracted from the existing data sets.”*

To address these points we analyzed existing data (see **Figure 2B** embedded above in response to Reviewer 1). In the reversibility data set, we further quantified animal locomotion by tracking the nose speed (**Fig. 2B**):

*“To test for reversibility, we used blue light (5 s pulse, 5 mW/cm²) as a strong stimulus and found that this rapidly and reliably reversed the quiescent state, leading to a dramatic increase in both behavioral activity and nose speed (**Figure 2B, Supplementary Movie 3**).” (pp. 8).*

We also included a brief note in the Methods that frame-by-frame subtraction is a more reliable method measure of sleep and wakefulness than posture:

“Not only is frame-by-frame subtraction less computationally expensive than other metrics such as posture or nose-tracking, but we found overall that this technique is less prone to the noise associated with small errors in tracking specific C. elegans body points.” (pp. 31).

In addition, we also included the average activity *before and after* mechanical stimulation in **Figure 2D** to show the baseline activity (see figure embedded above).

5. “Two pieces of literature that might help interpret the effects of mechanosensation on sleep amount: It has been suggested that ALA acts as a mechanoreceptor, could thus ALA mediate the effects of mechanical stimulation of sleep amount (Sanders et al 2013, PMID:24341457)? Also, mechanical stimulation has been shown to increase RIS activity (Spies et al. 2018, PMID: 29950594).”

We thank the reviewer for the suggestion and have added this literature as a potential explanation of our results and an area of future work:

“Many questions remain with respect to C. elegans sleep and the role of sensory inputs in driving sleep behavior. For example, sleep-promoting neurons such as ALA and RIS drive behavioral quiescence, but what ensemble activity occurs upstream of these circuits? The interaction between multiple sensory inputs, the ALA/RIS neurons, and diverse behavioral outputs are likely highly complex, non-linear systems involving multiple pathways and neuromodulators^{34,38,43–45,53}. Future work will help answer these questions and determine the relationship between ALA and RIS and whether these neurons are critical for driving sleep in all the environmental conditions tested in this report (Figure 6). Furthermore, ALA and RIS themselves are known to respond to mechanical stimulation^{86,89}, potentially providing a mechanism for why animal confinement drives sleep behavior. These “sleep-driving” neurons that also act as sensory neurons are a prime example of the multimodal nature of the C. elegans nervous system, which adds another element of complexity when dissecting the sensory and sleep circuits involved in environmentally-driven sleep.” (pp. 27-28).

6. “The paper emphasizes the behavioral criteria that define sleep, which include homeostasis. Traditionally, homeostasis is tested by sleep deprivation, which should result in an increased sleep amount or depth. This was first shown for C. elegans using sensory stimulation (Raizen et al. 2009). If the authors could provide such an experiment that would be helpful. For example, the worm could be kept awake for a defined period of time using blue light or mechanical stimulation as possible with the microfluidic devices developed by the authors and the subsequent sleep response could be quantified. A relatively simple experiment would seem sufficient to make the point.”

To address this point, we performed **new experiments** using optogenetic inhibition of the RIS neuron (see response to Editorial comments and **Supplementary Figure 2** embedded above). Although optogenetic inhibition of the RIS neuron appeared to have a relatively weak effect on the observed behavior during stimulation, we observed a significant increase in both sleep and low-activity behavior compared to a control group during the refractory period following stimulation. **These results suggest that animals show a homeostatic rebound in microfluidic-induced sleep following an extended period of RIS inhibition.**

REVIEWERS' COMMENTS:

Reviewer #1 (Remarks to the Author):

The authors have addressed my comments to a satisfactory level.

With respect to the new supplementary Figure 8 that addresses the other reviewer's concern, I think that the indicated periods should not be called "putative" sleep at all. With the information at hand these can be forward command states as well. Without IDs of forward-backward-turning-behavior related neurons and RIS, I would be much more careful with any conclusions.

Reviewer #2 (Remarks to the Author):

The authors have addressed my concerns. This study will be a valuable contribution to the field.

Reviewer #3 (Remarks to the Author):

The authors addressed most of my concerns. This is an interesting study that contains a lot of data on how microfluidics and sensory stimuli affect sleep behavior and this study will be valuable for the *C. elegans* neurobiology community. I recommend a minor revision and support publication. I don't need to see the manuscript again before it is published.

1) The point that was not addressed is the identification of neurons in their pan-neuronal imaging. Here it would be helpful to identify tentatively key players. In Figure 4 the forth trace from top could be RIS based activity, but spatial information is not given? Similarly, in their last supplementary figure in the second panel there is a trace that could be RIS. Based on their imaging data the authors could identify this trace tentatively as RIS. This is particularly important in the light of panel one of the last supplementary figure, which does not seem to have such a strong single trace during the quiescence period. This could be mentioned and discussed briefly in the figure legend. This could be added quickly without additional experimentation.

Others are really minor points:

2) Out of curiosity: why is the curvature increased during this type of microfluidic sleep? I don't expect the authors to know the answer, but I would like to read a sentence about why this might be the case.

3) The pan-neuronal movie still contains the term *uSleep*.

4) "while warmer temperatures promote sleep (Figure 6)." should read "while warmer temperatures promote sleep (Figure 6)."

We thank editor for another round of constructive criticism and the acceptance pending this revision. Overall these reviews have certainly strengthened the manuscript and we are looking forward to having it appear in Nature Communications. Below is a point-by-point response to the editor's suggested revisions, manuscript reformatting, and reviewer's comments.

EDITORIAL COMMENTS

1. *“We therefore invite you to revise your paper one last time to address the remaining concerns of our reviewers, in particular those of Reviewer 3.”*

In our response to the major point raised by Reviewer 3, we revised whole-brain imaging figures (Figure 4, Supplementary Figure 8), edited figure legends, and revised the text to hypothesize that some of the highly active neurons during sleep are likely the RIS interneuron.

2. *“At the same time we ask that you edit your manuscript to comply with our format requirements and to maximise the accessibility and therefore the impact of your work.”*

We accept the Editor's revisions to the abstract as well as the Editor's summary of our work. We also modified the manuscript title to “A Microfluidic-Induced *C. elegans* Sleep State” to remove any punctuation. In addition, we:

- a. Removed quotation marks unless directly quoting labels in a figure
- b. Included subheadings in the results section
- c. Shortened figure captions where necessary
- d. Shortened the main text to less than 5000 words
- e. Reduced references down to 70 citations

REVIEWER 1

Remarks to authors:

“The authors have addressed my comments to a satisfactory level. With respect to the new supplementary Figure 8 that addresses the other reviewer's concern, I think that the indicated periods should not be called "putative" sleep at all. With the information at hand these can be forward command states as well. Without IDs of forward-backward-turning-behavior related neurons and RIS, I would be much more careful with any conclusions.”

In the whole-brain imaging data heatmaps, we changed the label from “Putative sleep” to “State transition.” We also modified a sentence in the caption to read:

“Potential sleep states (top two panels) identified by a global state transition and downregulated neural activity, are labeled but cannot be confirmed without behavioral readouts.”
(Supplementary Figure 8).

REVIEWER 2

Remarks to authors:

“The authors have addressed my concerns. This study will be a valuable contribution to the field.”

REVIEWER 3

Remarks to authors:

“The authors addressed most of my concerns. This is an interesting study that contains a lot of data on how microfluidics and sensory stimuli affect sleep behavior and this study will be valuable for the C. elegans neurobiology community. I recommend a minor revision and support publication. I don’t need to see the manuscript again before it is published.”

Major points:

1. *“The point that was not addressed is the identification of neurons in their pan-neuronal imaging. Here it would be helpful to identify tentatively key players. In Figure 4 the forth trace from top could be RIS based activity, but spatial information is not given? Similarly, in their last supplementary figure in the second panel there is a trace that could be RIS. Based on their imaging data the authors could identify this trace tentatively as RIS. This is particularly important in the light of panel one of the last supplementary figure, which does not seem to have such a strong single trance during the quiescence period. This could be mentioned and discussed briefly in the figure legend. This could be added quickly without additional experimentation”*

In Figure 4, we added a scale bar and indicated the location of the neuron in the heatmap that increases in calcium activity. We also made a revision to the main text:

“However, as expected from previous work^{33,38,39,43,65}, we observed that some neurons actually increased in activity during microfluidic-induced sleep (Figure 4C), some of which corresponded with the approximate location of the RIS interneuron (indicated neuron in Figure 4A-B), which has been proposed as a sleep-promoting neuron for multiple C. elegans sleep-like states^{38,39,43}.”
(pp. 15-16).

We also added a micrograph to the whole-brain imaging data in Supplementary Figure 8 and indicated the location of a neuron that increases in calcium activity during the brain-state transition. We made the revision to the figure legend:

“In the top right heatmap, an indicated neuron that increases in activity during potential sleep corresponds to the approximate RIS location (circled neuron in top micrograph).”
(Supplementary Figure 8).

Minor points:

3. *“Out of curiosity: why is the curvature increased during this type of microfluidic sleep? I don’t expect the authors to know the answer, but I would like to read a sentence about why this might be the case.”*

To address this point we included the additional sentence:

*“This result is consistent with the fact that *C. elegans* quiescent posture, a hockey-stick-like shape^{57,59}, has a higher curvature than the long body wavelengths displayed during swimming locomotion⁴⁹.” (pp. 8).*

4. *“The pan-neuronal movie still contains the term uSleep.”*

We changed this label to just “Sleep” in the movie.

5. *“ ‘while warmer temperatures promote sleep (Figure 6).’ should read ‘while warmer temperatures promote sleep (Figure 6).’”*

We made this small edit to the text.